# TRIM25 predominately associates with anti-viral stress granules

Zehua Shang[1,7], Sitao Zhang[1,7], Jinrui Wang[1,7], Lili Zhou[2], Xinyue Zhang[1], Daniel D. Billadeau[3], Peiguo Yang [4], Lingqiang Zhang [5], Fangfang Zhou [2], Peng Bai[6] ✉ & Da Jia [1] ✉

Stress granules (SGs) are induced by various environmental stressors, resulting in their compositional and functional heterogeneity. SGs play a crucial role in the antiviral process, owing to their potent translational repressive effects and ability to trigger signal transduction; however, it is poorly understood how these antiviral SGs differ from SGs induced by other environmental stressors. Here we identify that TRIM25, a known driver of the ubiquitination-dependent antiviral innate immune response, is a potent and critical marker of the anti-viral SGs. TRIM25 undergoes liquid-liquid phase separation (LLPS) and co-condenses with the SG core protein G3BP1 in a dsRNA-dependent manner. The co-condensation of TRIM25 and G3BP1 results in a significant enhancement of TRIM25's ubiquitination activity towards multiple antiviral proteins, which are mainly located in SGs. This co-condensation is critical in activating the RIG-I signaling pathway, thus restraining RNA virus infection. Our studies provide a conceptual framework for better understanding the heterogeneity of stress granule components and their response to distinct environmental stressors.

Stress granules (SGs) are membraneless RNA granules formed in eukaryotic cells in response to various stresses, such as hypoxia, heat shock, viral infection, and oxidative stress[1]. The formation of SGs leads to the reprogramming of cellular metabolism, inhibition of translation, and alteration of normal cellular signaling to aid cell survival. SGs are formed through a physical process called liquid-liquid phase separation (LLPS), which contains many proteins including small ribosomal subunits, translation initiation factors, and translationally arrested mRNA and RNA-binding proteins[2]. A core SG network composed of ~36 SG proteins has been recently identified, with G3BP1/2 as the central node[3]. These proteins are critical for regulating the assembly

and functions of SGs. The importance of SG in human diseases is emphasized by the observation that dysregulation of SG assembly and disassembly occurs in many pathological conditions, such as viral infections[4–8], neurodegenerative diseases[9–11], and cancer[12,13].

Recent advances reveal that different types of stress induce distinct SG subtypes, which differ in their composition, dynamics of assembly and disassembly, and thus cellular functions[14]. For instance, sodium arsenite or heat shock treatment induces the formation of canonical SGs, which are more dynamic in nature and play a pro-survival role[15]. In contrast, Chemotherapy drugs like selenite or UV exposure led to the formation of noncanonical SGs, which are less

[1]Key Laboratory of Birth Defects and Related Diseases of Women and Children, Department of Pediatrics, West China Second University Hospital, State Key Laboratory of Biotherapy, Sichuan University, Chengdu 610041, China. [2]Institutes of Biology and Medical Science, Soochow University, Suzhou 215000, China. [3]Division of Oncology Research and Schulze Center for Novel Therapeutics, Mayo Clinic, Rochester, MN 55905, USA. [4]School of Life Sciences, Westlake University, Hangzhou 310024, 310030, China. [5]State Key Laboratory of Proteomics, Beijing Proteome Research Center, National Center for Protein Sciences (Beijing), Beijing Institute of Lifeomics, 100850 Beijing, China. [6]Department of Forensic Genetics, West China School of Basic Medical Sciences & Forensic Medicine, Sichuan University, Chengdu 610041, China. [7]These authors contributed equally: Zehua Shang, Sitao Zhang, Jinrui Wang. ✉e-mail: baipeng@scu.edu.cn; JiaDa@scu.edu.cn

dynamic and pro-apoptotic to cells[15,16]. Although the concept of stress-specific SG subtypes is emerging[7,15,17–19], their different protein and RNA compositions and functions are largely unexplored.

The formation of SGs is an integral part of the innate immune response to viral infection[7]. All viruses require the translation machinery of the host cell to synthesize viral proteins[5,20]. Accordingly, eukaryotic cells have evolved sophisticated mechanisms, including the formation of SGs, to shut down viral gene translation[5,20]. SGs can serve as a platform for the assembly of antiviral signaling pathways, such as the interferon response. For instance, many viral sensors, such as melanoma differentiation-associated Protein 5 (MDA5), retinoic acid-inducible gene I (RIG-I), and the double-stranded RNA-dependent protein kinase (PKR), are recruited to SGs to initiate antiviral responses[7]. The central proteins in SGs, G3BP1/2, promote both inflammatory and type I interferon (IFN) responses[21–23]. PKR and G3BP1 are involved in the positive regulation of innate immune sensors, such as RIG-I[8,21,24]. Despite this, viruses have developed sophisticated mechanisms to interfere with SG formation or to exploit SGs. For instance, Severe Acute Respiratory Syndrome Coronavirus 2 (SARS-CoV-2) utilizes its N protein to interact with G3BP1/2 and to inhibit the formation of SGs[25–27]. However, while numerous reports have linked SGs to viral infection[4–6,8], it is largely unknown how virus-induced SGs initiate the antiviral signaling pathways, and how these antiviral SGs differ from other types of SGs in terms of protein and nucleic acid content.

Here, we discovered TRIM25 as a potent marker of antiviral SGs, using a G3BP1 proximity-dependent biotinylation assay. We found that TRIM25 underwent LLPS, which was significantly enhanced by the presence of dsRNA. Both poly(I:C) treatment and infection with an RNA virus trigger the co-condensation of TRIM25 and G3BP1, which dramatically increased the ubiquitination activity of TRIM25 toward substrates, many of which were localized in SGs. The TRIM25 and G3BP1 co-condensation was critical in activating the RIG-I signaling pathway and restricting RNA virus infection. Our studies not only provide insights into the regulation of the antiviral signaling pathways, but establish a platform to investigate the composition, dynamics, and functions of stress-specific SG subtypes.

## Results

### RNA viruses or dsRNA mainly drive the recruitment of TRIM25 to stress granules

Previous studies have discovered that G3BP1/2 are essential for SG assembly under multiple conditions[3]. To identify specific SG components that respond to viral infection, we set to characterize the G3BP1-interaction network upon poly(I:C) stimulation using a proximity-dependent biotinylation (BioID) approach. Poly(I:C) is a synthetic analog of dsRNA, which is often used to mimic the action of RNA viruses[28]. We used the BirA* variant of an abortive biotin ligase, which provides higher labeling efficiency of proximal proteins[29] (Fig. 1a). Quantitative mass spectrometric analysis of the G3BP1-associated proteins upon poly(I:C) stimulation versus WT identified 937 differential hits out of 2906 biotinylated proteins (Figs. 1a, S1a, b and Data.S1). These hits were significantly enriched in multiple KEGG pathways, such as RIG-I-like receptor signaling pathway and viral or bacterial invasion (Fig. S1c). Among the 937 differential hits, 181 belonged to the previously identified SG proteins, including many SG core proteins such as HDAC6 and DDX3X (Fig. S1d, e and Data.S1). Interestingly, TRIM25, an established driver of the ubiquitination-dependent antiviral innate immune response, increased most dramatically among all the proteins and was enriched 130 times upon poly(I:C) treatment (Fig. 1b). The strong enrichment of TRIM25 in the SGs was intriguing, since the TRIM25 protein level only increased slightly over 2 folds in the same condition (Fig. S1f). This implies poly(I:C) treatment stimulates the recruitment of TRIM25 into the SGs (referred to as antiviral SGs).

To investigate whether the recruitment of TRIM25 into the SGs is specific to poly(I:C) treatment, we systematically treated HeLa cells that overexpressed GFP-TRIM25 with various types of stress inducers: RNA virus invasion (Sendai virus, SeV), exogenous dsRNA stress (poly(I:C), oxidative stress (sodium arsenite), ER stress (thapsigargin), translation inhibition (puromycin), proteasome inhibition (MG132), energy depletion (CCCP), heat shock, and osmotic stress (sorbitol) (Fig. 1c). As expected, all the inducers readily led to the formation G3BP1-positive granules. Intriguingly, while TRIM25 puncta were observed under multiple conditions, TRIM25 and G3BP1 puncta co-localized only when the cells were infected with SeV or treated with poly(I:C) (Fig. 1c, d). Similarly, TRIM25 exhibited the most significant interaction with G3BP1 upon poly(I:C) treatment in a proximity label-ing assay (Fig. S1g). On the other hand, osmotic stress or puromycin treatment did not increase the interaction between TRIM25 and G3BP1 relative to control. Furthermore, the interaction between TRIM25 and G3BP1 only slightly increased upon CCCP, MG132, sodium arsenite or thapsigargin treatment. Thus, the association between TRIM25 and G3BP1 is mainly driven by RNA viruses or dsRNA.

To test the dependence of G3BP1/2 in the formation of TRIM25 puncta, we generated cells with double knockout of G3BP1 and G3BP2 (referred to as G3BP dKO cells) and assessed the formation of TRIM25 puncta upon poly(I:C) stimulation. In WT U2OS cells, poly(I:C) treat-ment induced two types of G3BP1-positive puncta (Fig. 1e). The large puncta represented canonical SGs (Fig. 1e; middle panel), and the small ones likely corresponded to the RNase L–dependent bodies (RLBs) identified in previous studies[30,31] (Fig. 1e; lower panel). We also con-firmed the identity of the small puncta as RLBs by immunostaining with PABPC1 or TIA, protein makers used to distinguish RLBs from SGs. As expected for RLBs, these puncta co-stained with PABPC1, but not TIA. Similar to exogenously expressed TRIM25, endogenous TRIM25 formed puncta and exhibited strong co-localization with both types of G3BP1 puncta upon poly(I:C) treatment (Fig. 1e, f). We found that TRIM25 strongly co-localized with SGs and RLBs, but the co-localization with SGs was stronger than with RLBs. In G3BP dKO cells, TRIM25 still formed puncta upon poly(I:C) or sodium arsenite treatment, suggesting that the formation of TRIM25 puncta did not require G3BP1/2 (Fig. 1h). Relative to sodium arsenite treatment, poly(I:C) induced significantly fewer but larger TRIM25 puncta, sug-gesting that TRIM25 might have different functions under different types of stimuli. However, TRIM25 puncta had similar mean fluores-cence intensity in both conditions (Fig. 1k). In conclusion, our data suggest that G3BP1/2 are not necessary for the formation of TRIM25 puncta, but likely play a role in regulating the size or the number for TRIM25 puncta.

### dsRNA drives co-condensation of TRIM25 and G3BP1

To investigate whether TRIM25 foci form via LLPS, we used Fluores-cence Recovery after Photobleaching (FRAP) to probe the dynamic exchange of TRIM25 foci. GFP-TRIM25 was expressed in cells and poly(I:C) stimulation was added to induce dsRNA-associated TRIM25 foci. Photobleaching of GFP-TRIM25 condensates led to a fluorescence recovery within 60 s, reflecting a local exchange of TRIM25 molecules within the foci (Fig. 2a). To test whether TRIM25 could undergo LLPS in vitro, we expressed and purified MBP-tagged TRIM25, and labeled the protein with Alexa Fluor 488 dye. Tobacco etch virus protease (TEV) cleavage led to the formation of micrometer-scale liquid dro-plets in a time- and concentration-dependent manner (Figs. 2b and S2a). The formation of TRIM25 droplets was inhibited by high salt, suggesting that hydrophobic interactions likely play a role in the for-mation of the droplets (Fig. S2b). Using FRAP, we found that the dro-plets showed full recovery within 30 s after photobleaching, indicating a liquid state and fast exchange of materials with the environment (Fig. S2c). The addition of 56-bp dsRNA, which was previously shown to bind TRIM25 with high affinity[32], further increased the size and number

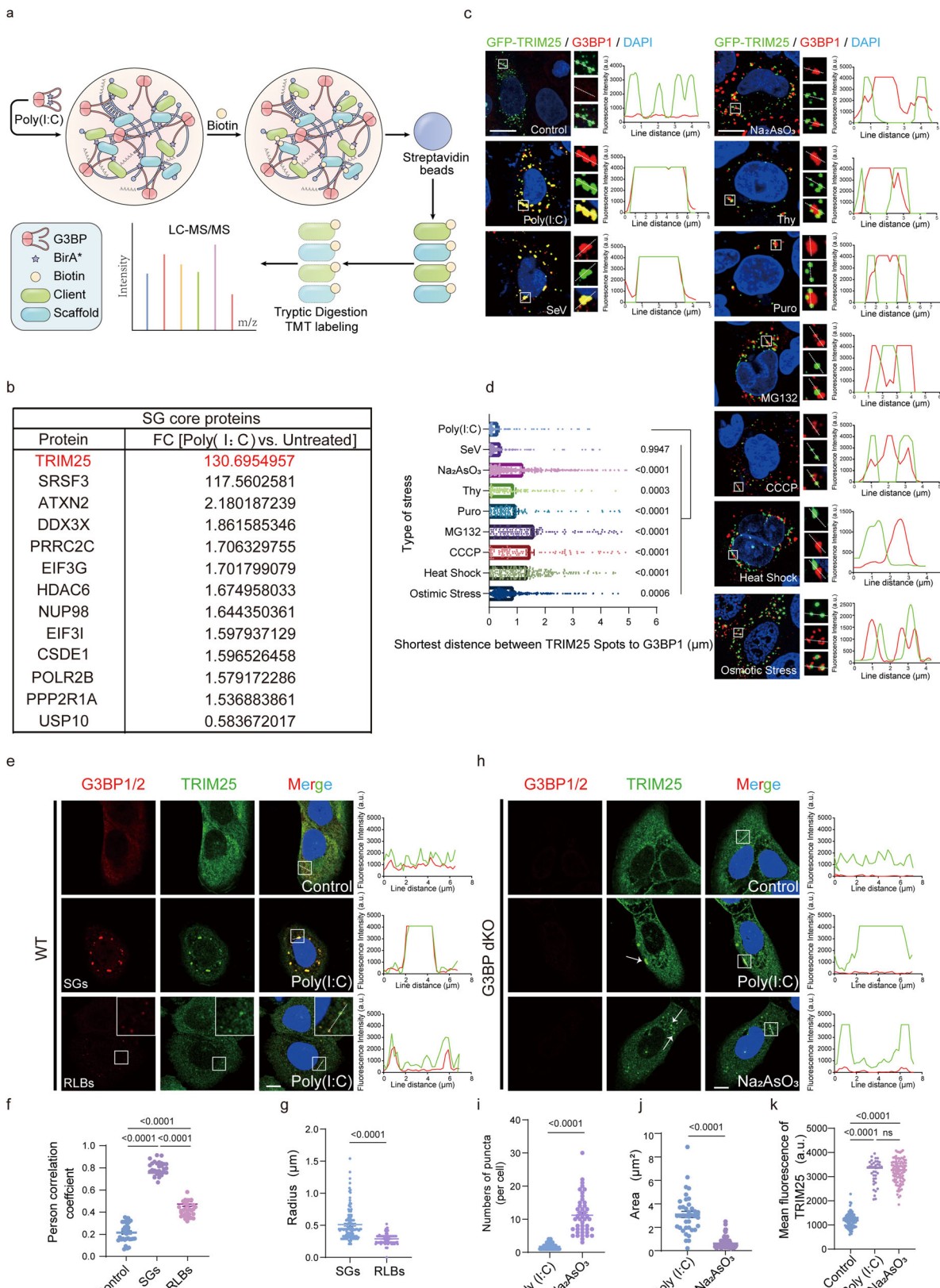

of TRIM25 droplets (Fig. S2d). Using fluorescence-labeled 56-bp dsRNA (cy5-dsRNA), we confirmed that TRIM25 formed co-condensation with dsRNA (Fig. S2e). To further characterize the TRIM25 condensates, we performed phase separation assays with various concentrations of protein and the 56-bp dsRNA. TRIM25 underwent phase separation at a concentration of 6.25 μM in the presence of 6.25 ng/μl dsRNA (Fig. 2c).

Thus, TRIM25 undergoes LLPS in vitro, which is further promoted by dsRNA.

Previous studies indicated that TRIM25 binds RNA with specific structures, and longer dsRNA (such as 28 bp and 56 bp) is more efficiently to enhance the ubiquitination activity of TRIM25 than shorter dsRNA (such as 14 bp)[33,34]. To determine how the dsRNA length

**Fig. 1 | The recruitment of TRIM25 to stress granules is predominantly driven by RNA viruses or dsRNA. a** Schematic depicting G3BP1 BioID methodology coupled to TMT-based quantitative proteomics. **b** A list of SGs core proteins induced by poly(I:C) treatment, as determined in (**a**). **c** Representative fluorescence microscopy images showing the colocalization of TRIM25 with endogenous G3BP1 in HeLa cell upon various types of stress. For induction conditions see "Immuno-fluorescence microscopy and live-cell imaging" section. Inset: higher magnification of the white boxed area. **d** The shortest distance between TRIM25 and G3BP1 foci in HeLa cells as in (**c**). Sev infection and Poly(I:C) treatment led to the shortest distance, among all types of stress. (Poly(I:C), $n = 205$; SeV, $n = 197$; Na$_2$AsO$_3$, $n = 403$; Thy, $n = 120$; Puro, $n = 114$; MG132, $n = 136$; CCCP, $n = 94$; Heat Shock, $n = 227$; Osmotic Stress, $n = 364$). **e−g** Formation of endogenous TRIM25 and G3BP1 puncta in WT U2OS cells. Cells were transfected with or without 2 μg/ml of poly(I:C) for 6 h

(**e**). Pearson's correlation coefficient showing significant co-localization of endogenous TRIM25 with G3BP1 in SGs and RLBs (Control, $n = 33$; SGs, $n = 31$; Control, $n = 34$). **f** Radius statistics for SGs and RLBs, SGs ($n = 128$) vs. RLBs ($n = 140$): $p$ value: 1.59e−021 (**g**). **h−k** Formation of endogenous TRIM25 puncta in G3BP dKO U2OS cells. White arrows indicate the TRIM25 puncta (**h**). Scatter plot of the numbers of TRIM25 puncta in G3BP dKO U2OS. Poly(I:C) ($n = 44$) vs. Na$_2$AsO$_3$ ($n = 44$): $p$ value: 2.51e−017 (**i**). Statistical analysis of TRIM25 droplet areas in (**h**), Poly(I:C) ($n = 39$) vs. Na$_2$AsO$_3$ ($n = 96$): $p$ value: 1.26e−027 (**j**). Fluorescence intensities of TRIM25 (Control, $n = 33$; Poly(I:C), $n = 31$; Na$_2$AsO$_3$, $n = 34$) (**k**). Data are representative of at least three independent experiments (**c, e, h**). Line scans show the related intensity profiles of TRIM25 and G3BP1. Scale bar, 10 μm (**c, e, h**). $n$ represents the number of samples. (a.u. arbitrary units.) Mean ± s.e.m., statistical analysis was performed using two-tailed Student's $t$ test (**g, i, j**) or one-way ANOVA (**d, f, k**).

impacted TRIM25 LLPS, we performed condensation experiments with purified TRIM25 protein and same mass of dsRNA of different lengths (14, 28, and 56 bp), and found that the 56 bp-dsRNA led to more TRIM25 droplets than the 14 bp- and 28 bp-dsRNA (Fig. S2f). To investigate the importance of RNA sequence in regulating TRIM25 LLPS, we included 56 bp-dsRNA$^{TATA}$, which was composed of multiple TA repeats, as a comparison. Interestingly, although both 56 bp-dsRNA and 56 bp-dsRNA$^{TATA}$ effectively mediated the formation of TRIM25 droplets (Fig. S2f), TRIM25 formed smaller droplets in the presence of 56 bp-dsRNA (Fig. S2g). Thus, the LLPS properties of TRIM25 are regulated by both the length and sequence features of dsRNA.

To illustrate how dsRNA regulate TRIM25 and G3BP1 condensation, we performed condensation assays with purified TRIM25 and G3BP1 proteins and the 56-bp dsRNA. Consistent with TRIM25 alone, cy5-dsRNA co-condensated with both TRIM25 and G3BP1 (Fig. S2h). Furthermore, cy5-dsRNA colocalized with TRIM25 and G3BP1 in cells (Fig. S2i). Although TRIM25 and G3BP1 could individually form droplets in the absence of dsRNA, the addition of dsRNA significantly sped up the process (Fig. 2d). The 56-bp dsRNA also increased both the TRIM25 intensity within the droplets and the number of TRIM25-G3BP1 co-condensate foci (Fig. 2e, f). Similar to TRIM25 alone, the presence of the 56-bp dsRNA led to more compacted TRIM25-G3BP1 co-condensate foci over time (Fig. 2g). To examine whether the dsRNA was specifically required for the co-condensation between TRIM25 and G3BP1, we tested other types of nucleic acids with the same length. Remarkably, dsRNA, but not other nucleic acids (ssRNA, ssDNA, dsDNA), efficiently promoted the fusion of TRIM25 and G3BP1 droplets (Fig. 2h, i). Thus, dsRNA specifically promotes TRIM25-G3BP1 co-condensation.

Next, we asked whether TRIM25 could alter the LLPS property of G3BP1. We found the addition of TRIM25 reduced the protein and RNA concentration required for the formation of G3BP1 droplets, indicating that TRIM25 facilitated the LLPS of G3BP1 (Fig. 2j, k). Using FRAP assays, we found that G3BP1 recovered more quickly when mixing with TRIM25 than on its own (Fig. 2l). These observations suggested that TRIM25 promotes G3BP1-mediated LLPS and condensate dynamics.

### The direct interaction between TRIM25 and G3BP1 is required for their co-condensation

So far, our data show that TRIM25 and G3BP1 undergo dsRNA-dependent co-condensation. Next, we sought to determine whether TRIM25 directly interacts with G3BP1 and how the interaction contributes to the co-condensation. Using an antibody against TRIM25, we were able to detect the interaction between endogenous TRIM25 and G3BP1 (Fig. S3a). Interestingly, poly(I:C) treatment did not further enhance the interaction (Fig. S3b). To map the TRIM25 region responsible for G3BP1 binding, we generated a series of truncations and mutations of TRIM25 (Fig. 3a). TRIM25 harbors a "404-PTFG-407" motif between its coiled-coil and PRY/SPRY domains, which displays some degree of similarity with the previously identified G3BP-binding motif, ΦxFG$^{35–38}$ (where Φ is a hydrophobic residue and X is any

residue). Indeed, both alanine substitution (PTFG$^{AAAA}$) and deletion of the PTFG motif (ΔPTFG) abolished the interaction between TRIM25 and G3BP1 in immunoprecipitation (IP) experiments, like previous reports$^{36}$ (Fig. 3b, c). Intriguingly, single or double replacement of the four amino acids in the PTFG motif to alanine abolished the interaction with G3BP1, highlighting the importance of the integrity of the PTFG sequence (Fig. S3c). In addition to the PTFG motif, deletions of the PRY/SPRY, coiled-coil, or the 7 KA mutant dramatically impaired the ability of TRIM25 to immunoprecipitate G3BP1 (Fig. 3b, c). Both the PRY/SPRY and 7K regions of TRIM25 have been previously shown to contact dsRNA$^{32,33,39}$, implying that the binding of TRIM25 to dsRNA promotes interactions with G3BP1.

To determine how the TRIM25-G3BP1 interaction contributes to the co-condensation, mCh-TRIM25 (WT or ΔPTFG) and GFP-G3BP1 were transiently co-transfected into HeLa cells, and TRIM25 foci and SGs dynamics were visualized using live-cell imaging. TRIM25 WT and G3BP1 foci appeared less than 5 h after poly(I:C) transfection, and displayed strong co-localization (Fig. 3d). In contrast, the formation of TRIM25 ΔPTFG foci was significantly delayed and was not observed until 8 h after poly(I:C) treatment (Fig. 3d). Furthermore, the co-condensation frequency between TRIM25 ΔPTFG and G3BP1 was much less than that between TRIM25 WT and G3BP1 (Fig. 3d). These observations were further supported by immunofluorescence experiments, which showed that SGs and TRIM25 ΔPTFG formed separated condensates upon treatment of sodium arsenate or poly(I:C) (Fig. 3e). Thus, the PTFG motif is required for TRIM25 and G3BP1 interaction and co-condensation.

To further support our conclusion, we performed LLPS experiments using purified proteins. The recombinant MBP-TRIM25 ΔPTFG and MBP-G3BP1 proteins were labeled with different fluorescent dyes and incubated with TEV in physiological salt conditions at room temperature. TEV cleavage led to the formation of individual liquid droplets of TRIM25 and G3BP1 (Fig. 3f). We also mixed TRIM25 and G3BP1 solutions, in which individual droplets were performed. Interestingly, although the mixed proteins were half of their original concentrations due to the dilution effect, TRIM25 WT and G3BP1 droplets still rapidly fused with each other and formed co-condensate droplets that were much larger than the individual droplets (Fig. 3g, h). Like TRIM25 WT, TRIM25 ΔPTFG also formed micrometer-scale condensates in a concentration-dependent manner (Fig. S2a). The number and size of TRIM25 ΔPTFG condensates increased with protein concentrations (Fig. S2a). However, unlike TRIM25 WT, TRIM25 ΔPTFG and G3BP1 droplets did not fuse with each other after remixing (Fig. 3i). And upon mixing the solutions, the size of both droplets decreased, rather than increased, likely due to a dilution effect (Fig. 3j). Concomitantly, the number of TRIM25 ΔPTFG and G3BP1 individual droplets greatly increased relative to that of TRIM25 WT and G3BP1 (Fig. 3k). Thus, the PTFG motif of TRIM25 is necessary for the co-condensation between TRIM25 and G3BP1.

We and others have previously discovered that the SARS-CoV-2 N protein utilizes an ITFG motif to contact G3BP1/2 to facilitate viral

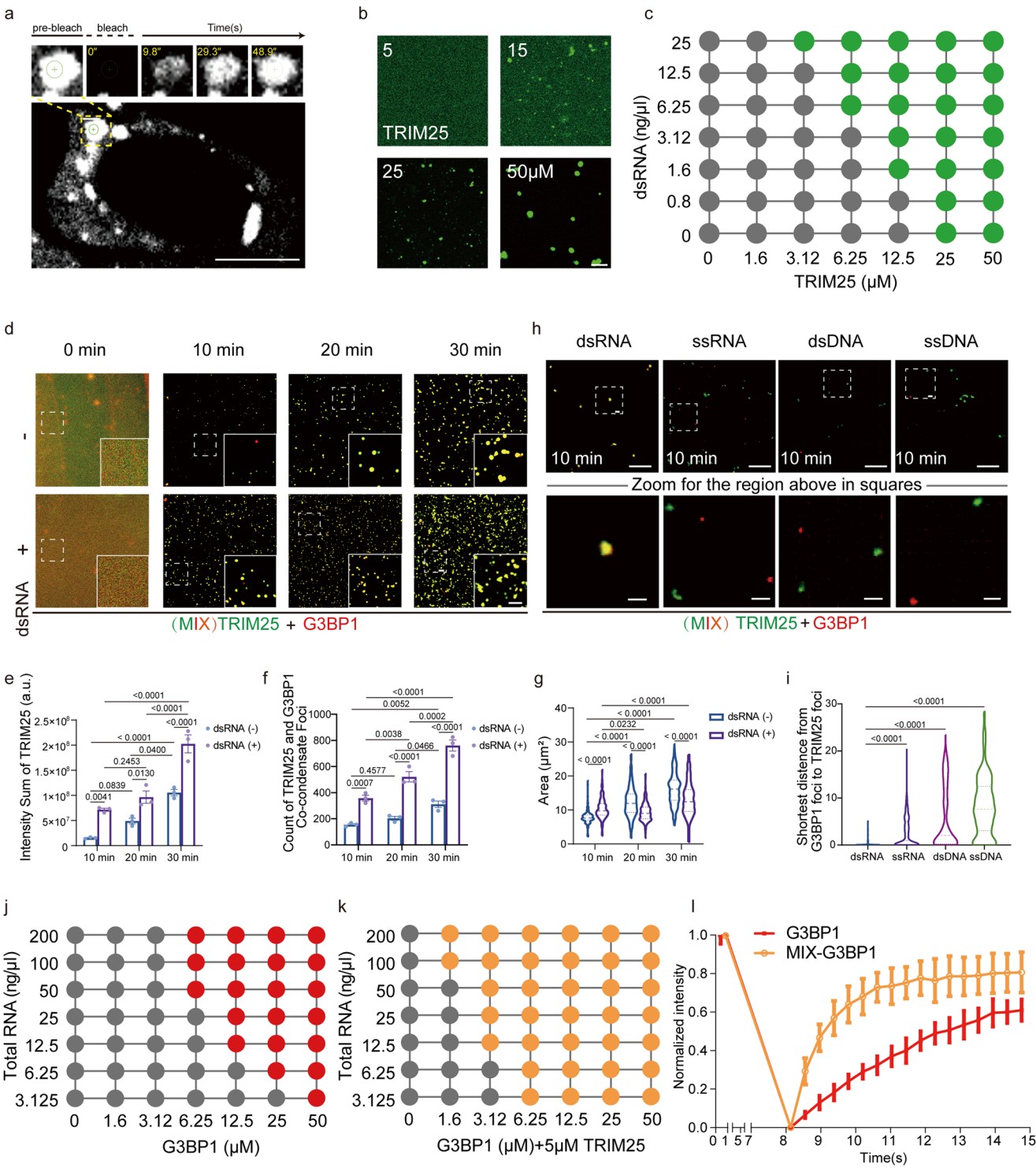

**Nature Communications** | (2024)15:4127

infection[25,38]. The similarity between the ITFG motif and the PTFG motif in TRIM25 suggests a competitive binding to G3BP1. Indeed, we found that the SARS-CoV-2 N protein could strongly compete with the TRIM25-G3BP1 interaction (Fig. S3d). Isothermal Titration Calorimetry (ITC) experiments demonstrated that a peptide harboring the N protein ITFG motif bound to G3BP1 with a stoichiometry of 1:1 and with an affinity of KD = 7.5 ± 0.7 μM (Fig. S3e). This affinity was slightly higher than that between the TRIM25 PTFG motif and G3BP1 (KD = 12.7 ± 1.2 μM) (Fig. S3e). In contrast, no binding was detected when the PTFG motif was removed (Fig. S3e). Furthermore, SARS-CoV-2 N proteins also disrupted the TRIM25 and G3BP1 co-condensation (Fig. S3f), confirming that the TRIM25-G3BP1 interaction is required for their co-

condensation. Interestingly, the H1N1 NS1 protein also disrupted the TRIM25 and G3BP1 co-condensation (Fig. S3f), which likely functioned via interfering with the formation of the TRIM25 dimer[40].

## Co-condensation between TRIM25 and G3BP1 facilitates TRIM25 to access SG-localized antiviral proteins

To understand whether and how TRIM25 exerts its antiviral functions via SGs, we employed a combination of TRIM25-BirA*-based proximity labeling with quantitative mass spectrometry (MS) (Figs. 4a and S4a). By comparing the TRIM25 WT and ΔPTFG interactomes, we identified 1061 differential hits out of 2160 biotinylated proteins (Fig. 4b and Data.S1). Among these hits, 209 proteins belonged to the previously

**Fig. 2 | dsRNA drives co-condensation of TRIM25 and G3BP1. a** FRAP recovery image of GFP-TRIM25 droplets in HEK293T cells upon poly(I:C) stimulation. **b** Representative fluorescence images of purified TRIM25 at various concentrations. **c** Summary of in vitro LLPS behavior of purified TRIM25 and a 56-bp dsRNA. Green: LLPS; gray: no LLPS. **d**–**g** Time-dependent formation of TRIM25-G3BP1 LLPS in the absence (top) or presence (bottom) of the 56-bp dsRNA (2 ng/μl). Fluorescence microscopy images of 20 μM TRIM25 and 20 μM G3BP1 mixture (**d**). Sum intensity in (**d**), dsRNA (−): 10 min vs. 20 min, p value: 6.90e−002; 20 min vs. 30 min, p value: 3.52e−002. dsRNA (−) vs. dsRNA (+): 10 min, p value: 4.12e−003; 20 min, p value: 1.30e−002; 30 min, p value: 3.44e−005 (**e**). Counts in (**d**), dsRNA (−): 10 min vs. 20 min, p value: 0.4577; 20 min vs. 30 min, p value: 4.66e−002. dsRNA (−) vs. dsRNA (+): 10 min, p value: 7.46e−004; 20 min, p value: 1.12e−005; 30 min, p value: 2.73e−007. **f** Areas in (**d**), dsRNA (−): 10 min vs. 20 min, p value: <0.0001; 20 min vs.

30 min, p value: <0.0001. dsRNA (−) vs. dsRNA (+): 10 min, p value: 6.92e−010; 20 min, p value: 6e−014; 30 min, p value: <0.0001 (**g**). **h, i** Only the dsRNA induced the merge of G3BP1 and TRIM25 droplets in 10 min after the addition of nucleic acids (2 ng/μl). **h** Fluorescence microscopy images of 20 μM TRIM25 and 20 μM G3BP1 mixture. Lower panel: zoomed images. **i** The shortest distance between G3BP1 and TRIM25 showing significant co-localization of two droplets upon the addition of dsRNA. **j, k** Summary of the LLPS behaviors of purified recombinant G3BP1 and total RNA, in the presence or absence of TRIM25. Red (**j**) and yellow (**k**): LLPS; gray: no LLPS. **l** Graphic presentation of fluorescence recovery dynamics after photobleaching of areas inside the droplets. All data are representative of at least three independent experiments (**a**–**c, d, h, j**–**l**). Scale bar, 10 μm; Scale bar in Zoom, 2 μm (**a, b, d, h**). Mean ± s.e.m., statistical analysis was performed using one-way ANOVA (**i**) or two-way ANOVA (**e**–**g**).

identified SG proteins. Interestingly, 60% of the SGs proteins (128 out of 209) were enriched in the TRIM25 WT sample relative to ΔPTFG, confirming that the PTFG motif is critical for the association between the TRIM25 condensates and SGs (Fig. S4b).

In addition, we also compared the TRIM25 WT interactomes upon poly(I:C) treatment versus untreated. Two hundred sixteen out of 1178 differential hits belonged to the SG proteins, and a large proportion of them (69 out of 216) were found to be enriched upon poly(I:C) treatment (Fig. S4b). Analysis using DAVID bioinformatics resources revealed that differential hits of TRIM25 WT versus ΔPTFG, or TRIM25 WT poly(I:C) treatment versus untreated, share several common pathways, including SGs assembly and disassembly, activation of the innate immune response, and inhibition of viral replication (Fig. S3a, b).

To gain insight into how TRIM25 and SGs cooperate to exert antiviral functions, we manually curated a list of 134 annotated Antiviral Proteins (AVPs) using criteria of MS/MS count >9 in our mass spectrometry data (Fig. S4c and Data.S1). Fifty-six out of 134 AVPs were enriched in the TRIM25 WT sample versus ΔPTFG (Fig. 4c). Notably, a large percent of them (19 out of 56) have been reported to function in regulating cellular immune and/or antiviral responses together with TRIM25, including Zinc-Finger Protein ZCCHC3[41,42], and m6A reader IGF2BP2[43] (Fig. 4c). Remarkably, a comparison of the TRIM25 WT poly(I:C) vs. untreated proximal interactomes revealed a similar pattern, including ZCCHC3, IGF2BP2, and Zinc-Finger Antiviral Protein ZC3HAV1 (also known as ZAP[44–48]) (Fig. 4d). These data indicate that TRIM25 and G3BP1 may co-condensate to facilitate the antiviral functions and innate immune responses upon viral invasion.

To test whether viral infection promotes the co-localization of TRIM25 and these AVPs, we examined the subcellular localizations of G3BP1, TRIM25 and multiple AVPs in TRIM25 KO HeLa cells that were rescued with GFP-TRIM25 WT or ΔPTFG. We also included RIG-I in our experiment, which is known to translocate to the TRIM25-associated SGs upon SeV infection[49]. Like RIG-I, ZC3HAV1, ZCCHC3, and IGF2BP2 displayed strong co-localization with both G3BP1 and TRIM25 upon poly(I:C) treatment, confirming our mass spectrometric results (Fig. 4e). In cells expressing GFP-TRIM25 ΔPTFG, RIG-I and ZCCHC3 showed pronounced co-localization with G3BP1, but not with TRIM25 ΔPTFG, indicating that these AVPs were recruited to SGs independent of TRIM25 (Fig. 4e). IGF2BP2 showed significant co-localization with both G3BP1 and TRIM25 ΔPTFG foci, consistent with its association with both proteins (Fig. 4e). Unlike them, ZC3HAV1 co-localized with TRIM25 ΔPTFG foci, but not with poly(I:C)-induced SGs, likely reflecting a tight interaction between ZC3HAV1 and TRIM25[44–48] (Fig. 4e). We also examined the subcellular localization of AVPs with GFP-TRIM25 condensates in G3BP dKO HeLa cells. RIG-I and ZCCHC3 did not co-localize with GFP-TRIM25; in contrast, IGF2BP2 and ZC3HAV1 displayed strong co-localization with TRIM25 foci (Fig. S4e). Taken together, our data indicate that the co-condensation between TRIM25 and G3BP1 may provide a compartment to regulate the functions of proteins critical for innate immunity, such as RIG-I[50–53], ZC3HAV1[41,44–48],

ZCCHC3[41,42], and IGF2BP2[43]. Within the compartment, TRIM25 and G3BP1 could regulate the localization and functions of these proteins via diverse mechanisms.

## Co-condensation between TRIM25 and G3BP1 promotes the ubiquitination activity of TRIM25 towards AVPs

Since TRIM25 is an E3 ubiquitin ligase that mediates the K63-linked polyubiquitination[54–58], we next examined whether TRIM25 was responsible for polyubiquitination of proteins localized within SGs. HepG2 cells were treated with poly(I:C), and then immunostained with antibodies against G3BP1 and K63-polyUb (Fig. S6a). In WT cells, G3BP1 and K63-polyUb displayed strong co-localization. Deletion of TRIM25 dramatically diminished the co-localization, indicating that TRIM25 was largely responsible for the K63-linked polyubiquitination of SG proteins (Fig. S6a). Importantly, the diminished co-location between G3BP1 and K63-polyUb could be rescued by the re-expression of TRIM25 WT, but not by TRIM25 ΔPTFG (Fig. 6a), emphasizing the importance of TRIM25-G3BP1 interaction in regulating the ubiquitination of SG proteins.

We previously showed that TRIM25 and G3BP1 only co-condensate when the cells are treated with poly(I:C), but not with sodium arsenite (Fig. 1c). Thus, we wondered whether poly(I:C) treatment could enhance the ubiquitination activity of TRIM25 towards AVPs. HepG2 cells were treated with poly(I:C) or sodium arsenite, and the ubiquitination level of endogenous IGF2BP2 was determined (Fig. S6b). We chose IGF2BP2 as its available antibody could be used for the IP experiment. Immunoblotting analysis revealed that poly(I:C) treatment led to a higher ubiquitination level of IGF2BP2 relative to sodium arsenite treatment (Fig. S6b). Deletion of TRIM25 markedly diminished the ubiquitination level of endogenous IGF2BP2 in HEK293T cells treated with poly(I:C) (Fig. 5b). Importantly, the decreased ubiquitination level of IGF2BP2 could be rescued by re-expressing of TRIM25 WT, but not by that of TRIM25 ΔPTFG or CS mutants (Fig. 5b). To further demonstrate the importance of co-condensation with G3BP1 in regulating the ubiquitination activities of TRIM25, we transfected plasmids encoding ZC3HAV1, ubiquitin and TRIM25 in WT or G3BP1 KO cells and measured the ubiquitination levels of ZC3HAV1. We chose G3BP1 KO cells, rather than G3BP dKO cells, in these experiment as the former seemed much healthier than the latter. Poly(I:C) treatment significantly enhanced ubiquitination of ZC3HAV1 mediated by TRIM25 WT, but not by TRIM25 ΔPTFG. Deletion of G3BP1 dramatically reduced TRIM25 WT-mediated ubiquitination of ZC3HAV1 (Fig. 5c). To further confirm our results, we enriched ubiquitinated proteins using tandem-ubiquitin binding entity (TUBE) followed by immunoblotting (Fig. S6c). Poly(I:C) stimulation increased the ubiquitination levels of RIG-I, IGF2BP2, and PABPC1, another TRIM25 substrate. Importantly, the ubiquitination levels of the substrates were further increased when TRIM25 WT, but not TRIM25 ΔPTFG, was expressed. On the other hand, G3BP1 deletion diminished the substrate ubiquitination levels (Fig. S6c). Thus, TRIM25 and G3BP1 cooperate to regulate the ubiquitination of AVPs upon poly(I:C) treatment.

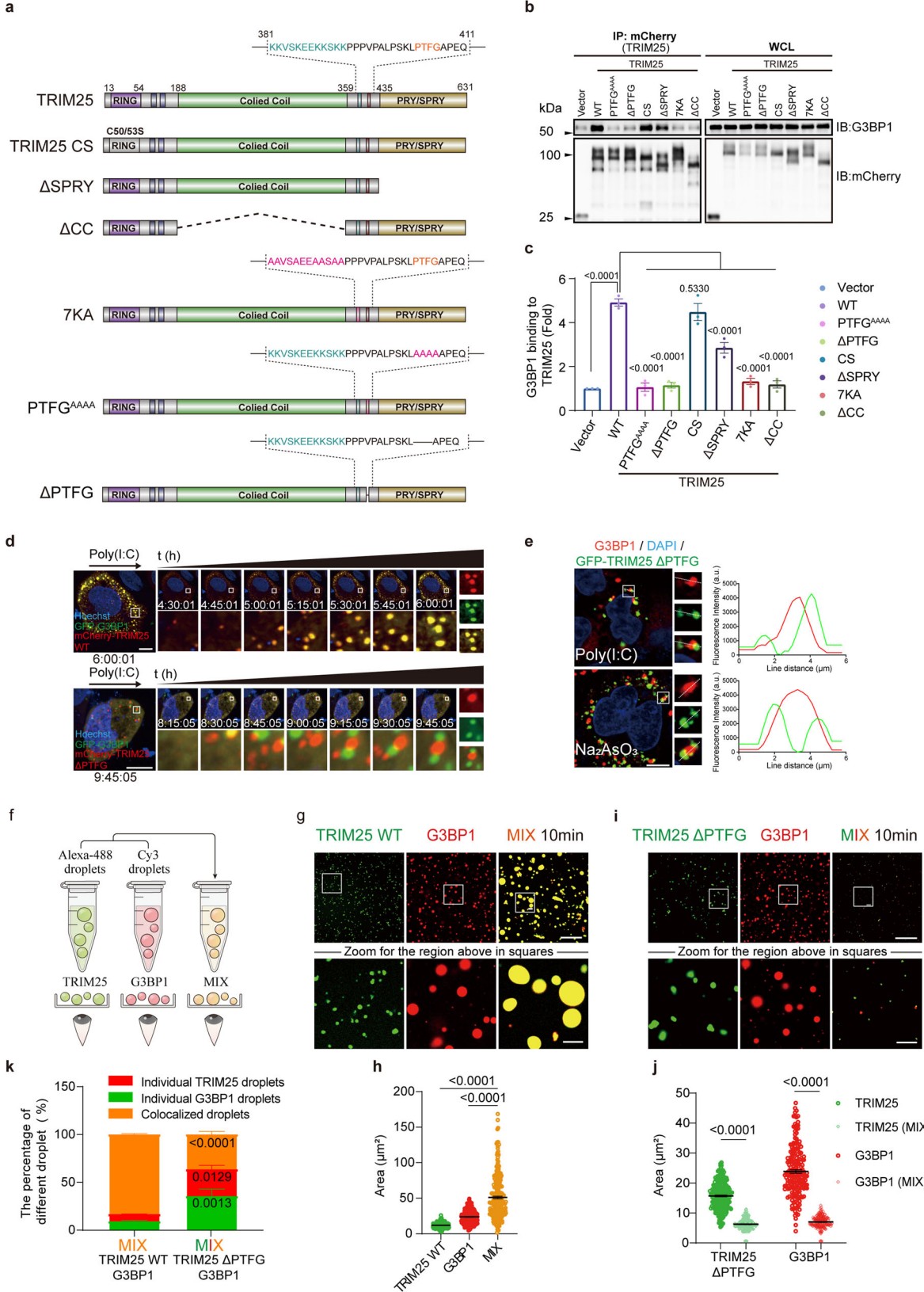

To characterize whether the co-condensation with G3BP1 regulated the ubiquitination activities of TRIM25 in vitro, we reconstituted the ubiquitination reaction. UbcH5b is known E2 ubiquitin-conjugating enzymes that function together with TRIM25. Accordingly, we found that full-length TRIM25 efficiently synthesizes polyUb with UbcH5b (Fig. S6d). The addition of G3BP1 further enhanced the production of polyUb in a concentration-dependent manner (Fig. S6d). Consistent with our LLPS experiment, we found that the 56 bp-dsRNA$^{TATA}$ did not enhance TRIM25 E3 ligase activity as efficiently as the 56 bp-dsRNA, indicating that dsRNA regulates TRIM25 E3 ligase activity via altering the LLPS property of TRIM25 (Fig. S6e).

**Fig. 3 | The direct interaction between TRIM25 and G3BP1 is required for their co-condensation. a** Schematic diagram of full-length (FL) and various truncations/point mutations of TRIM25 used in the experiment. **b** IP experiments to assess the interaction between TRIM25 and G3BP1. HEK293T cells were transfected with mCherry-TRIM25 FL, the indicated TRIM25 mutants, or empty vector for 24 h. The cell lysates were subjected to immunoprecipitation with mCherry beads followed by immunoblotting with indicated antibodies. **c** Quantification of immunoblots. **d** A time-lapse micrograph of G3BP1 (green) and TRIM25 WT or ΔPTFG (red) speck formation after poly(I:C) transfection, and the zoomed images of the speck at 6 or 10 h after transfection. Inset: higher magnification of the white boxed area. **e** The PTFG motif of TRIM25 is required for co-localization with G3BP1 in HeLa cells. Experiments were performed similar to those in Fig. 1c, except that the cells were transfected with GFP-TRIM25 ΔPTFG, instead of TRIM25 WT. Line scans show the related intensity profiles of TRIM25 and G3BP1. **f** Experimental workflow for two-color droplet mixing assay. **g, h** Fluorescence microscopy images of 50 μM TRIM25 droplets, 50 μM G3BP1 droplets, or their mixture (with 50 ng/μl total RNA). Quantitative analyses of droplet areas in (**g**). For TRIM25, $n = 372$; G3BP1, $n = 201$; MIX: TRIM25 + G3BP1, $n = 259$; $n$ represents the number of droplets (**h**). **i, j** Fluorescence microscopy images of 50 μM TRIM25 ΔPTFG droplets, 50 μM G3BP1 droplets, or their mixture. Quantitative analyses of droplet areas in (**g**). For TRIM25, $n = 256$; TRIM25 (MIX), $n = 192$; G3BP1, $n = 186$; G3BP1 (MIX), $n = 179$; $n$ represents the number of droplets (**j**). **k** Quantitative analyses of the percentage of different droplets after mixing G3BP1 and TRIM25 in (**h**) and (**j**). TRIM25 WT + G3BP1 vs. TRIM25 ΔPTFG + G3BP1: TRIM25 droplets, $p$ value: 3.25e−004; G3BP1 droplets, $p$ value: 3.59e−003; colocalized droplets, $p$ value: 1.94e−007. Data are representative of at least three independent experiments (**b–e, j, i**). Scale bars, 10 μm; Scale bars in Zoom, 2 μm (**d, e, j, i**). (a.u. arbitrary units.) Mean ± s.e.m., statistical analysis was performed using one-way ANOVA (**c, h**) or two-way ANOVA (**j, k**).

We next used purified RIG-I-2CARD-HA (amino acids 1-227) and ZC3HAV1-N-HA (amino acids 1-202) as substrates since both of them were reported substrates of TRIM25[47,52,59]. Similar to previous studies[56], unanchored polyUb chains were synthesized in the presence of E1 ligase, Ubc13/UbE2V2 (E2 ligase) and TRIM25 (E3 ligase). Unlike Ubc13/UbE2V2, UbcH5b (E2 ligase) allowed Ub chains to be anchored to the substrates. G3BP1 further enhanced the activity of TRIM25 WT, as assessed by the ubiquitination levels of RIG-I-2CARD and ZC3HAV1-N. Consistent with the competitive binding between TRIM25 and SARS-CoV-2 N protein toward G3BP1, the addition of the ITFG peptide from SARS-CoV-2 inhibited TRIM25 activity (Fig. 5d, e), Thus, G3BP1 may promote TRIM25 ubiquitination activity via co-condensation.

To further confirm the importance of G3BP1 in promoting the ubiquitination activity of TRIM25, we assessed two different G3BP1 mutants (TBM: TRIM25-binding mutant and GDM: G3BP1-dimer mutant) in the in vitro ubiquitination assay. We and others have previously identified three bulky hydrophobic amino acids (F15, F33, F124) in the NTF2 domain of G3BP1 that were critical for binding to the ΦxFG motif-containing peptide[25,60], and the TBM mutant (F15A, F33A, F124A) disrupted the TRIM25-G3BP1 interaction. Similarly, structural analysis revealed three amino acids (D81, H83, Q93) responsible for the homodimerization of G3BP1 (Fig. S6f), and the GDM mutant (D81K, H83A, Q93A) were expected to weaken the dimerization. Like G3BP1 WT, G3BP1 TBM underwent LLPS in the same concentration. In contrast, G3BP1 GDM was unable to form LLPS in vitro, confirming the importance of multivalency for LLPS (Fig. 5f). Interestingly, both G3BP1 TBM and GDM did not efficiently form SGs compared with G3BP1 WT in G3BP dKO cells, indicating that both G3BP1 dimerization and interaction with other proteins were necessary for the formation of SGs in cells (Fig. S6g). Unlike G3BP1 WT, G3BP1 TBM did not further enhance TRIM25 activity (Fig. S6d). Meanwhile, compared to G3BP1 WT, both G3BP1 TBM and GDM mutants failed to enhance the TRIM25-dependent ubiquitination of ZC3HAV1-N and RIG-I-2CARD (Fig. 5g, h). Thus, G3BP1, via self-dimerizing and interacting with TRIM25, enhances TRIM25 ubiquitination activity.

### Co-condensation between TRIM25 and G3BP1 activates the RIG-I signaling pathway

TRIM25 is known to regulate RIG-I ubiquitination, promote the phosphorylation of IRF3 and IRF7, and subsequently activate the transcription of type I interferon (IFN) genes in response to viral infection[52,56,57]. To investigate whether TRIM25-mediated RIG-I ubiquitination depends on G3BP1, we compared TRIM25, PTFG^AAAA, and ΔPTFG. Cells were transfected with plasmids encoding RIG-I, ubiquitin and TRIM25 WT or mutants, and then subjected to poly(I:C) treatment. TRIM25 WT significantly promoted the ubiquitination of RIG-I, like previous studies[52,56,57]. In contrast, TRIM25 PTFG^AAAA or ΔPTFG failed to enhance the ubiquitination (Fig. 6a). Consistently, deletion of G3BP1 diminished TRIM25-mediated RIG-I ubiquitination (Fig. 6b). Thus, our results indicate that TRIM25 and G3BP1 cooperate to regulate the ubiquitination of RIG-I.

Next, we tested whether TRIM25 promoted the phosphorylation of IRF3 via co-condensation with G3BP1. HEK293T cells were transfected with TRIM25 WT or mutants and then activated with poly(I:C). Ectopic expression of TRIM25 WT significantly increased the phosphorylation level of IRF3 at Ser396 (p-IRF3) upon poly(I:C) treatment, although it did not alter the protein level of IRF3. Compared to TRIM25 WT, the G3BP1-binding mutants TRIM25 PTFG^AAAA and ΔPTFG did not increase the level of p-IRF3 (Fig. 6c), suggesting that TRIM25 activated IRF3 via interacting with G3BP1. We found that deletion of G3BP1 decreased overall p-IRF3 levels upon poly(I:C) treatment. Interestingly, TRIM25 could enhance the IRF3-p level in both WT and G3BP1 KO cells (Fig. 6d). Furthermore, quantitative RT-PCR (qPCR) assays showed that deletion of G3BP1 also decreased poly(I:C)-induced gene expression of IFNβ and ISG56 (interferon-stimulated gene 56) (Fig. 6e, f).

To further confirm the relevance of TRIM25-G3BP1 condensation in regulating the type I IFN signaling, we determined the RNA levels of multiple critical genes in the pathway using quantitative RT-PCR (qPCR). TRIM25 WT significantly enhanced the expression of IFNα, IFNβ, IFNγ, and ISG56 upon poly(I:C) treatment. In contrast, the expression of these IRF3-dependent genes was abolished when TRIM25 PTFG^AAAA or ΔPTFG was transfected (Fig. 6g–j). Taken together, our data indicate that co-condensation between TRIM25 and G3BP1 is critical for regulating the RIG-I signaling pathway.

### Co-condensation between TRIM25 and G3BP1 resists SeV virus replication

Given the importance of TRIM25-G3BP1 co-condensation in regulating the RIG-I signaling pathway, we next tested whether this co-condensation was critical for viral infection. TRIM25 KO HEK293T cells were transfected with empty vector, TRIM25 WT or PTFG^AAAA, and then infected with equal amount of SeV (MOI = 1). We measured the amount of viral RNA at 18 h post infection (hpi) and found that deletion of TRIM25 increased SeV replication compared to wild-type HEK293T cells. Importantly, SeV replication could be inhibited by re-expression of TRIM25 WT, but not by that of the empty vector. TRIM25 PTFG^AAAA did not, efficiently inhibited SeV replication as TRIM25 WT, suggesting that TRIM25 inhibited SeV replication via G3BP1 (Fig. 7a). Furthermore, we quantified the expression of type I IFN and related genes in the presence of SeV infection. Deletion of TRIM25 resulted in decreased expression of IFN-β, ISG56, and CXCL10 compared to wild-type HEK293T cells. Re-expression of TRIM25 WT, but not PTFG^AAAA, significantly promoted the expression of these genes (Fig. 7b–d). Together, these results suggest that viral infection promotes TRIM25 co-condensation with G3BP1, which in turn inhibits viral infection.

## Discussion

The formation of SGs emerges as an important anti-stress mechanism in response to diverse environmental insults[9,13,20]. How SGs selectively

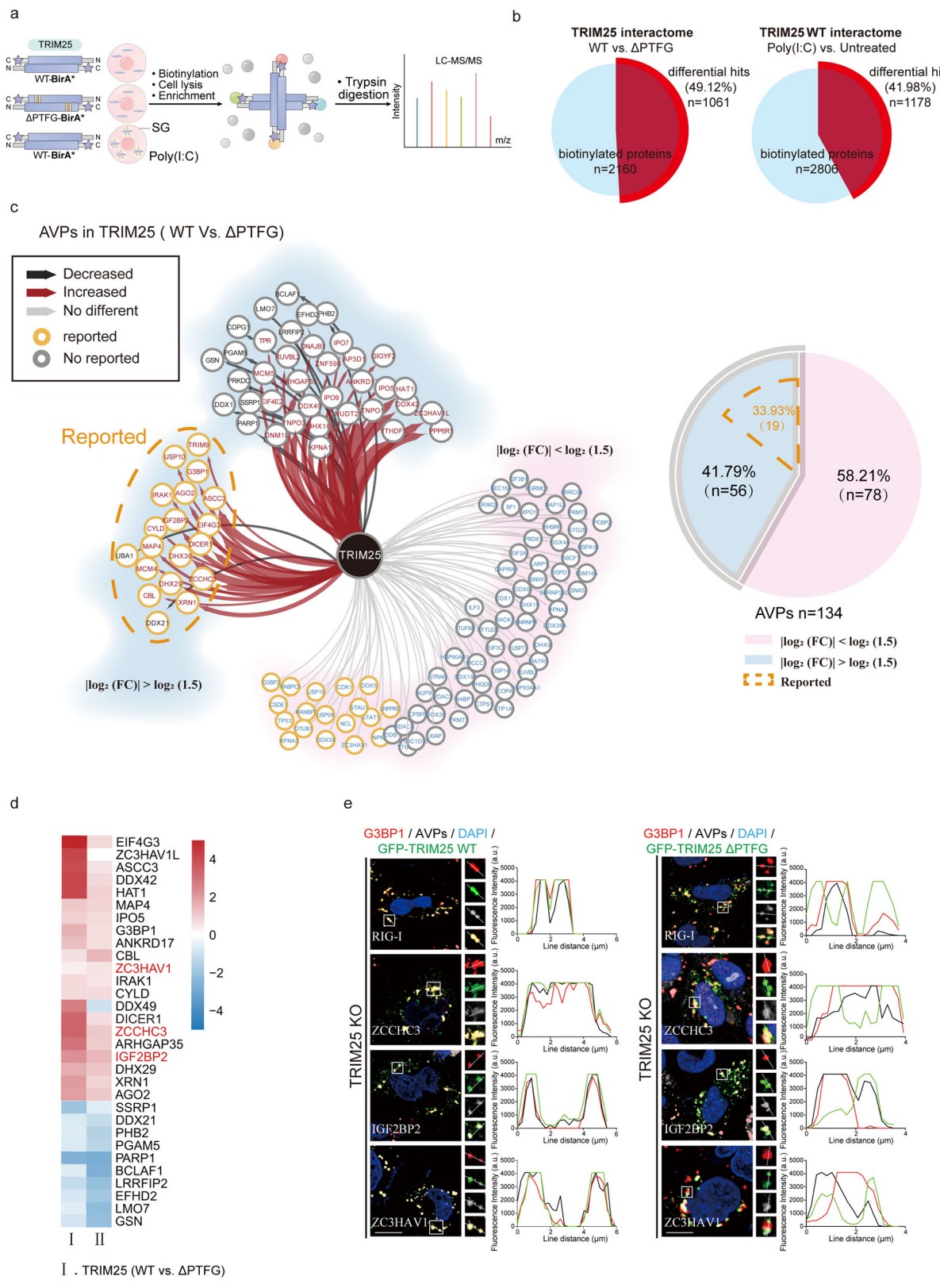

d.

Ⅰ. TRIM25 (WT vs. ΔPTFG)

Ⅱ. TRIM25 WT [Poly(I:C) vs. Untreated]

recruit specific proteins to elicit stress-specific responses has remained elusive. In this study, we discovered that TRIM25 predominately associated with antiviral SGs (Fig. 7f). TRIM25 directly interacted with G3BP1 and underwent co-condensates in a dsRNA-dependent manner. Importantly, the formation of TRIM25 antiviral SGs was mainly stimulated by RNA virus infection or poly(I:C) treatment. Furthermore, the TRIM25 antiviral SGs provided a compartment for enriching numerous AVPs and functioned in regulating multiple antiviral signaling pathways, such as the RIG-I-MAVS pathway. Together, our studies set up a paradigm to characterize the compositional and functional heterogeneity of SGs induced under various stress-inducing conditions.

**Fig. 4 | Co-condensation with G3BP1 promotes TRIM25 to access SG-localized AVPs. a** Schematic depicting TRIM25 BioID methodology coupled to TMT-based quantitative proteomics. **b** Summary of comparison of TRIM25 WT vs. ΔPTFG or poly(I:C) vs. untreated proximal interactors. **c** TRIM25 WT vs. ΔPTFG differential proximal interactors from 134 AVPs. Cytoscape software was used to generate network (left) representing significance analysis of INTeractome (SAINT)-filtered putative TRIM25-prey differential interactions identified by BirA*. Each node represents a selected hit from the screen, organized by the presence or absence of differences, and the relationship between increases and decreases. Label those proteins that have been reported to synergize immunomodulation with TRIM25. Pie chart (right) is a broad generalization of the TRIM25 WT vs. ΔPTFG differential proximal interactors from AVPs. **d** Comparison of TRIM25 WT vs. ΔPTFG or TRIM25 WT poly(I:C) vs. untreated proximal interactors reveals a remarkable concordance in the change of AVPs. **e** Representative images of AVPs subcellular localization HeLa cells. Cells were transfected with GFP-TRIM25 WT or ΔPTFG and then treated with poly(I:C). DAPI staining is shown in blue. Line scans show the related intensity profiles of TRIM25, G3BP1 and AVPs. Data are representative of at least three independent experiments (**e**). Scale bars, 10 μm (**e**). (a.u. arbitrary units.).

How do antiviral SGs differ from SGs induced by other types of stress? We show that the formation of the TRIM25 antiviral SGs is primarily promoted by dsRNA and TRIM25, whose expression is also triggered by RNA virus infection. The antiviral SGs promote the ubiquitination activity of TRIM25 towards AVPs likely through multiple mechanisms: (1) bringing TRIM25 and its substrates together and enriching them in the dense phase; (2) enhancing the enzymatic activity of TRIM25 through co-condensation with G3BP1; (3) altering the condensate dynamics, as shown in the in vitro experiment (Fig. 2h, i). Altogether, the inclusion of TRIM25 brings not only antiviral SGs unique compositions but also distinct physical properties and cellular functions. Recently, it is reported that TRIM25 can ubiquitinate and stabilize G3BP1[61]. It is consistent with our results that TRIM25 is a critical regulator of SGs.

The TRIM25 antiviral SGs could regulate the antiviral signaling pathways in multiple ways. First, ZC3HAV1 plays a critical role in the host response to viral infection and functions by promoting the degradation of viral mRNAs[44,62]. The SG localization of ZC3HAV1 is critical for regulating its antiviral activity[46]. We showed that abrogating the interaction between TRIM25 and G3BP1 disrupted the recruitment of ZC3HAV1 to the SGs, suggesting that TRIM25 recruits ZC3HAV1 to the SGs to regulate its activities. We further showed that the formation of antiviral SGs, via the co-condensation between TRIM25 and G3BP1, greatly enhanced the E3 ligase activity of TRIM25 toward ZC3HAV1 and other substrates. Second, our studies demonstrated that TRIM25 activated the RIG-I signaling by forming TRIM25-G3BP1 co-condensation and uncovered stress granules as key subcellular compartments in regulating the RIG-I pathway. It remains controversial whether TRIM25 directly catalyzes the ubiquitination of RIG-I[52,53,58]. We found that ZCCHC3, a protein that can bridge RIG-I and TRIM25, was recruited to the TRIM25 antiviral SGs upon poly(I:C) treatment[41], suggesting that ZCCHC3 might facilitate the ubiquitination of RIG-I by TRIM25 in SGs. Alternatively, antiviral SGs could allow the local enrichment of TRIM25-catalyzed unanchored K63-ubiquitin chains, which facilities the activation of the RIG-I pathway[58]. Lastly, our proteomic studies have identified dozens of AVPs, which could be regulated by TRIM25 antiviral SGs. Remarkably, only a small portion of them have been linked with TRIM25 in previous studies. Future studies will be necessary to address how TRIM25, stress granules, and these AVPs coordinately confer effective antiviral responses.

In addition to diverse environment insults, genetic mutations, aging, and other chronic stresses could also induce the formation of pathological SGs[9,63–65]. As a consequence, disturbance in SG dynamics has been regarded as a major driver for many neurodegenerative diseases, such as amyotrophic lateral sclerosis (ALS) and frontotemporal dementia (FTD)[9,63,64]. Moreover, a recent study reveals that diverse Charcot-Marie-Tooth type 2 neuropathies (CMT2)-causing mutants share the same properties by abnormally entering the SG network[66]. This perturbs the property of "normal" SGs, and disrupts SG-mediated stress responses. Very recently, Sur et al. demonstrate that SGs could function as "shock absorbers" to prevent excessive innate immune responses[67]. By identifying TRIM25 as the critical component for the antiviral stress granules, we anticipate that the framework established in our study could be used to investigate other SGs, irrespective of the types of stress. Further research will be needed to fully understand the diversity of stress granules, how they are regulated, and how they contribute to cellular stress responses in different contexts. Ultimately, this knowledge could help to develop more targeted therapies for stress-related diseases.

# Methods
## Antibodies
Antibodies against G3BP1 (66486-1-lg), G3BP2 (16276-1-AP), TRIM25 (12573-1-AP), IRF3 (11312-1-AP), IGF2BP2 (11601-1-AP), ZC3HAV1 (66413-1-lg), and RIG-I (67556-1-lg), GFP (50430-2-AP), mCherry (26765-1-AP), HA (51064-2-AP), GAPDH (10494-1-AP), Flag (20543-1-AP) were from Proteintech. The ZCCHC3 (A17235) was from ABclonal. The Ub (GT7811) was from GeneTex. IRF3 (phosphor Ser396) (ABP54922) was from Abbkine. Goat anti-Rabbit IgG-HRP (L3012) and Goat anti-Mouse IgG-HRP (L3032) were from Signalway Antibody (SAB). Goat anti-Rabbit IgG (H + L) Cross-Adsorbed Secondary Antibody, Alexa Fluor 546 (A-11010), Goat anti-Mouse IgG (H + L) Cross-Adsorbed Secondary Antibody, Alexa Fluor 546 (A-11003), Goat anti-Mouse IgG (H + L) Cross-Adsorbed Secondary Antibody, Alexa Fluor 488 (A-11001), Goat anti-Rabbit IgG (H + L) Cross-Adsorbed Secondary Antibody, and Alexa Fluor 488 (A-11008) were from Thermo Fisher. DAPI (Servicebio, CR2110134) and Hochest (Yeasen, 40732ES03) were purchased for use as a cell nuclear dye.

## Chemicals
The following compounds were used to treat cells: Poly(I:C) LMW (InvivoGen, tlrl-picw-250), Sodium Arsenite (Merck, 7784-46-5), CCCP (Selleck, S6494), Thapsigargin (TargetMol, TQ0302), Puromycin (Sangon, A610593-0026), MG132 (TargetMol, T2154), Sorbitol (TargetMol, T0332).

## Plasmids
Human TRIM25 and G3BP1 were cloned into pcDNA3.1(+), mCherry-C1, or pEGFPC1. ZC3HAV1, SARS-CoV-2 N, H1N1 NS1, and RIG-I were cloned into pcDNA3.1(+) or mCherry-C1. All the mutations were generated using High-Fidelity PCR kit (MACLAB), and the sequences were confirmed by DNA sequencing.

## Cell culture and transfection
HEK293T, HepG2, U2OS and HeLa cells were purchased from ATCC and were cultured in DMEM medium containing 10% fetal bovine serum, 100 U/ml penicillin, 100 U/ml streptomycin. The cells were grown in an incubator kept at 37 °C and 5% $CO_2$. When the cells reached 80% fusion by apposition growth, trypsin was added for digestion and passage. HEK293T cells and Hela cells were transfected at ~80% confluence with Hieff Trans™ Liposomal Transfection Reagent (Yeasen, 40802ES03) according to the manufacturer's instructions.

## Immunoprecipitation and immunoblotting
Lysates were prepared using the same procedures as when performing immunoprecipitation. Lysates were incubated with GST-TUBEs-beads at 4 °C overnight. Beads were washed three times with ice-cold PBS,

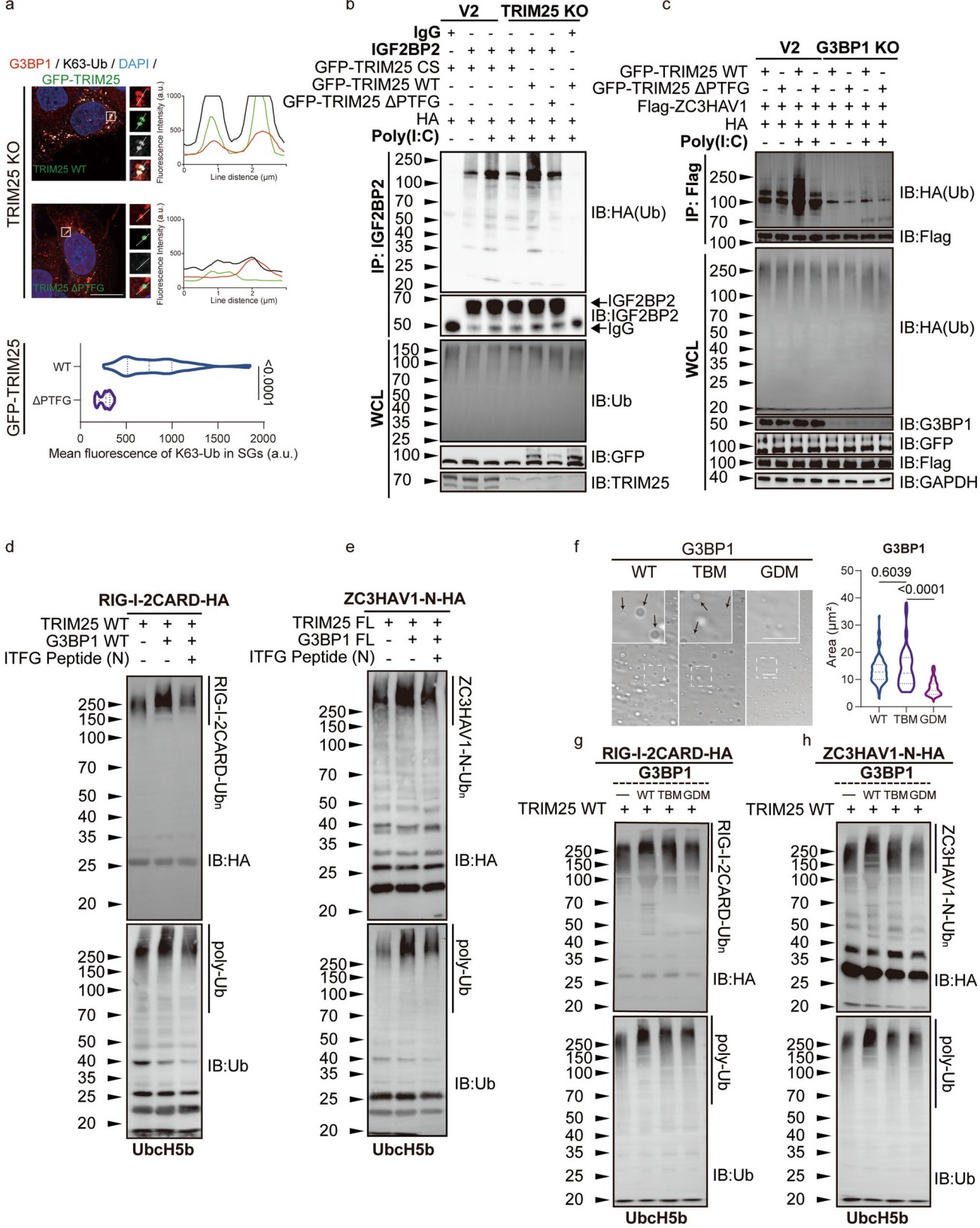

mixed with the SDS loading buffer, and boiled at 90 °C for 5 min. The resulting samples were analyzed by immunoblotting.

## BirA*-mediated biotinylation and proteomics

BirA*-mediated biotinylation experiments were performed similar to previous studies[68]. HEK293T cells were transfected with plasmids encoding G3BP1-BirA* and TRIM25-BirA* or TRIM25-ΔPTFG-BirA*. Twelve hours after plasmid transfection, the cells were treated with a 20 ug/ml poly(I:C) at 37 °C for 10 h. At the same time, biotin was added to the medium to a final concentration of 500 μM. The cells were washed once with cold PBS, and lysed in a cocktail lysis buffer (8 M urea, 20 mM Tris-HCl, pH 7.4, 150 mM NaCl, 1% protease

**Fig. 5 | Co-condensation with G3BP1 enhances ubiquitination activity of TRIM25 toward various AVPs. a** Immunofluorescent staining of G3BP1 and K63-linked ubiquitin in TRIM25 KO HeLa cells re-expressing GFP-TRIM25 WT or ΔPTFG. Line scans show the related intensity profiles of GFP-TRIM25, G3BP1 and Ub. Bottom: Fluorescence intensities of and K63-linked ubiquitin in G3BP1-positive SGs. TRIM25 WT vs. TRIM25 ΔPTFG: $p$ value: 1.89e−042; (**b**) WT or TRIM25 KO HEK293T WT cells were transfected with indicated GFP-TRIM25 constructs, and treated with or without poly(I:C). The cell lysates were subjected to immunoprecipitation with IgG or IGF2BP2 antibody followed by immunoblotting with indicated antibodies. **c** WT or G3BP1 KO HEK293T cells were transfected with the indicated plasmids and treated with or without poly(I:C). Anti-Flag immunoprecipitates were analyzed by immunoblot. **d, e** In vitro ubiquitination analysis of RIG-I-2CARD-HA and ZC3HAV1-N-HA. Assays contained E1 ligase, UbcH5b (E2 ligase), TRIM25 (E3 ligase), the 56-bp dsRNA, Ub, PBS, ITFG peptide (30 μM), and RIG-I-2CARD-HA (**d**) or ZC3HAV1-N-HA

(**e**). Poly-Ub chain was analyzed by anti-Ub blot. Ubiquitination of RIG-I-2CARD-HA and ZC3HAV1-N-HA was analyzed by anti-HA blot. **f** Formation of 50 μM G3BP1 WT or mutant liquid droplets visualized by DIC microscopy. TBM TRIM25-bingding mutant, GDM G3BP1-dimmer mutant. Black arrows indicate the G3BP1 droplets. Quantitative analyses of G3BP1 droplet areas in (**f**). RIG-I-2CARD (**g**) and ZC3HAV1-N (**h**) in vitro ubiquitination assays with indicated G3BP1 mutants (10 μM) and E1/E2 (UbcH5b)/TRIM25. Assays contained purified E1 ligase, UbcH5b (E2 ligase), TRIM25 (E3 ligase), G3BP1 mutants (WT; TBM: TRIM25-binding mutant; GDM: G3BP1-dimmer mutant), the 56-bp dsRNA, Ub, RIG-I-2CARD-HA (**d**) or ZC3HAV1-N-HA (**e**). WT panels show corresponding control experiments, performed in parallel. Samples were analyzed by IB with anti-Ub and anti-HA. All data are representative of at least three independent experiments (**a**–**h**). Scale bars, 10 μm (**a**). (a.u. arbitrary units.) Mean ± s.e.m., statistical analysis was performed using two-tailed Student's $t$ test (**a**) or one-way ANOVA (**f**).

inhibitor). The lysate was centrifuged at $15,000 \times g$ for 10 min. For biotin pull-down experiments, the supernatant was incubated with streptavidin beads (200 μl) overnight at 4 °C. The streptavidin beads were washed three times with the lysis buffer. A portion of the sample was boiled in the SDS loading buffer at 100 °C for 5 min and analyzed by immunoblotting. The remaining samples were used for quantitative mass spectrometry analysis, similar to previous studies[69]. The results of the mass spectrometry analyses were listed in Supplementary Data 1.

### Immunofluorescence microscopy and live-cell imaging
Immunofluorescence experiments were performed as previously reported[70,71]. HeLa cells were grown on 14 mm diameter coverslips, and was induced with various inducers: SeV (MOI = 0.1) for 12 h, poly(I:C) (2 μg/ml) for 10 h, 2 mM sodium arsenite for 0.5 h, 10 μM thapsigargin for 1 h, 20 μg/ml puromycin for 3 h, 10 μM MG132 for 3 h, 20 μM CCCP in glucose-free medium for 1 h. The cells were fixed with 4% paraformaldehyde, and incubated with the appropriate antibodies. Images were acquired by Olympus FV3000 and analyzed using Olympus confocal software.

For live-cell imaging experiments, HeLa cells were transfected with plasmids encoding mCherry-TRIM25 WT or ΔPTFG and GFP-G3BP1. The cells were cultured on 24-well glass-bottom dishes for 1 day, and then stimulated with Poly(I:C). Images were acquired using a Spinning Disk Confocal microscope (Olympus SpinSR10) with 405 nm, 488 nm, and 546 nm lasers for Hoechst, GFP, and mCherry signals, respectively

### Establishment of KO cell lines using CRISPR–Cas9 genome editing
G3BP dKO and G3BP1 KO cells were established by a plenti-CRISPR/Cas9-v2 system[72]. The sequences of target G3BP1/2-sgRNAs were 5′-CACCGTGTCCGTAGACTGCATCTGC-3′ (G3BP1) and 5′-CACCG-TACTTTGCTGAATAAAGCTC-3′ (G3BP2), respectively. The V2 vector with puromycin resistance was transiently transfected into HEK293T and HeLa cells. Forty-eight hours after transfection, cells were screened for 3 days using puromycin (2 μg/ml). Single cell clones were isolated by dilution and expanded in culture, and G3BP dKO cells were screened by immunoblotting. TRIM25 KO cells were established similarly, and were described in previous studies[73].

### FRAP experiments
FRAP experiments were performed on an Olympus FV3000 confocal microscope system, similar to previous studies[66,74]. For in vitro protein droplets, droplets of ~2 μm diameter were photobleached using 488 nm and 546 nm lasers at around 10% laser power and maintained for 4 s. After bleaching, time-lapse images were obtained at ~2 min intervals of about 1 s.

For cellular droplets, FRAP experiments were performed in a live-cell imaging chamber at 37 °C. HeLa cells were transfected with the

pEGFP-TRIM25 plasmid for 6 h and then treated with poly(I:C) for 6 h. Complete or partial photobleaching of the target droplets was achieved using a laser at 488 nm with about 10% laser power for 4 s. After bleaching, time-lapse images were obtained over a time course of 1.5 min.

The fluorescence intensity of the region of interest (ROI) above was corrected by the unbleached control region and then normalized to the pre-bleached intensity of the ROI. The corrected and normalized data were fitted to a single exponential model using the OlyVIA software developed by Olympus.

### RT-qPCR
Total RNA was extracted from HEK293T using an Easy Pure® RNA Kit (TransGen, Beijing, China), and was performed according to the manufacturer's instruction. The 1st strand cDNA was synthesized by TransScript First-Strand cDNA Synthesis Super Mix (TransGen). The cDNA was subjected to real-time PCR using SYBR Green Master (Roche, Basel, Switzerland) as previously described[75]. The results were normalized by GAPDH. Primers used for these analyses were listed in Supplementary Table S1.

### TUBE assay
Lysates were prepared using the same procedures as when performing immunoprecipitation. Lysates were incubated with GST-TUBEs-beads at 4 °C overnight. Beads were washed three times with ice-cold PBS, mixed with the SDS loading buffer, and boiled at 90 °C for 5 min. The resulting samples were analyzed by immunoblotting[76].

### Sendai virus infection
Sendai virus (SeV) infection experiments were performed similar to previous studies[77]. In brief, SeV was diluted with serum-free DMEM and incubated with HEK293T cells. After incubation for 1 h, the medium was replaced by DMEM containing 10% FBS.

### Protein expression and purification
G3BP1 WT and variants were expressed and purified as previously described[25]. Briefly, human G3BP1 was cloned into pMAL–based expression vector (NEB), encoding a N-terminal MBP tag followed by a TEV cleavage site. The cells were lysed by a high-pressure homogenizer, and cleared by centrifugation at $20,000 \times g$. The cleared supernatant was passed through amylose beads (NEB), and the bound proteins were eluted in a buffer containing 20 mM maltose. The proteins were further purified by ion exchange with Q column. The eluted proteins were then subjected to Superdex 200 Increase 10/300 (GE Healthcare) chromatography.

Full-length TRIM25 was cloned into a pFastBac vector (Thermo Fisher) with an N-terminal 10×His or 10×His MBP tag. Proteins were expressed in SF9 insect cells, similar to previous studies[40]. Briefly, the cells were resuspended in buffer (50 mM Tris, pH 8.0, 20 mM NaCl, 10% (v/v) glycerol, 1.5% (v/v) Triton X-100, 1 mM TCEP). The cells were

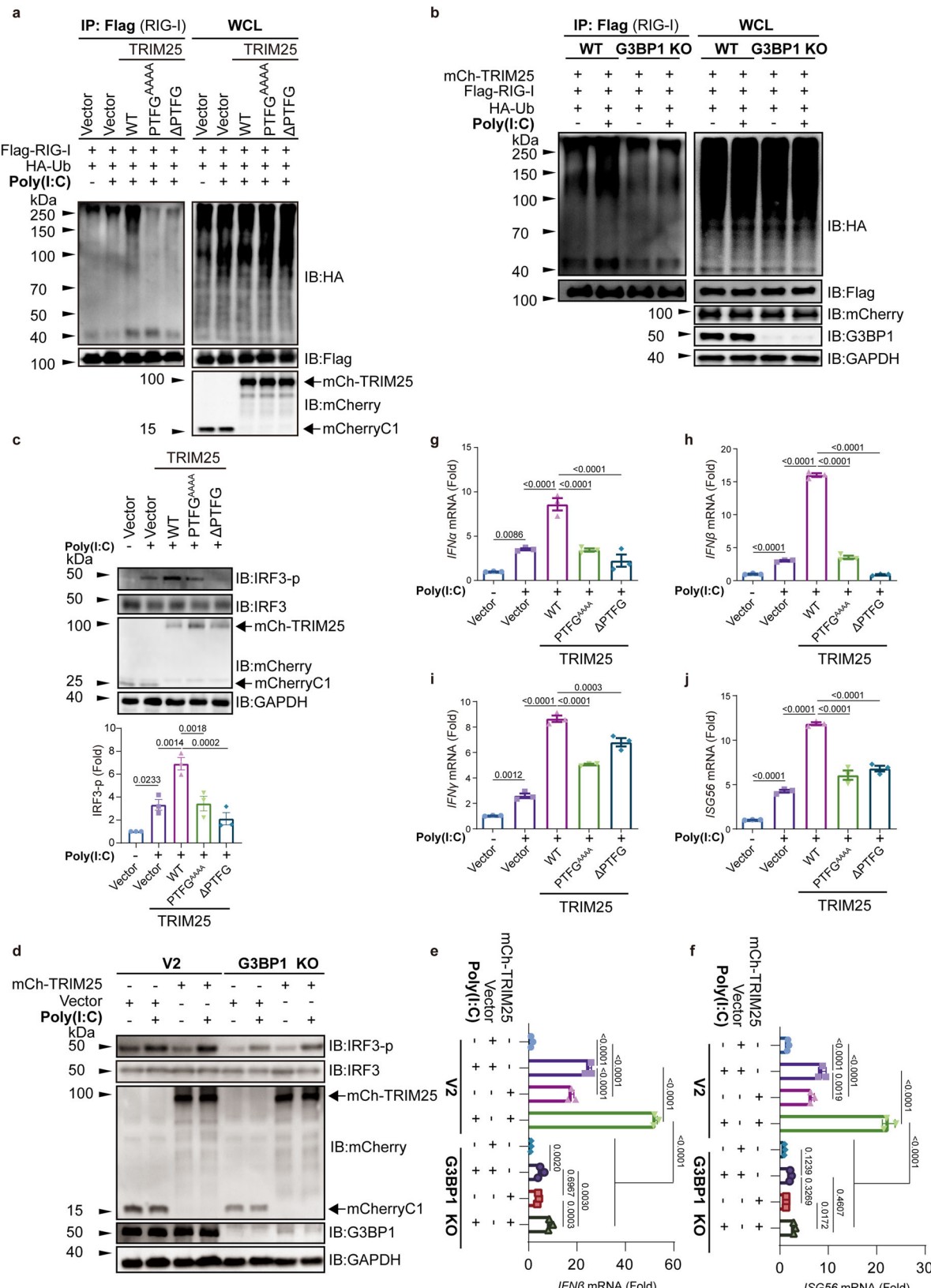

lysed by ultrasonication, and the debris was removed by centrifugation. The His-tagged TRIM25 was first purified by a Ni-NTA column (Qiagen), and followed by Superdex 200 Increase 10/300 column (GE Healthcare).

The genes encoding ZC3HAV1-N and RIG-I-2CARD inserted into the pET28a vector with an N-terminal 6×His tag and a C-terminal HA tag. Plasmids were transformed into E. coli BL21(DE3) cells, and bacteria were grown in LB medium at 37 °C until the OD600 reached 0.6. Subsequently, the proteins were induced by IPTG and expressed at 18 °C for 16 h. The proteins were purified by Ni2 + -NTA and gel filtration chromatography, and finally eluted in a buffer containing 20 mM Tris-HCl, pH 7.5, and 150 mM NaCl.

**Fig. 6 | Co-condensation of TRIM25 and G3BP1 activates the RIG-I signaling pathway. a** HEK293T cells were transfected with HA-Ub, Flag-RIG-I plasmids, together with mCherry-TRIM25 (mCh-TRIM25) and its indicated mutant plasmids. After poly(I:C) treatment, anti-Flag immunoprecipitates were analyzed by immunoblot with HA, Flag, and mCherry antibodies. **b** HEK293T WT or G3BP1 KO cells were transfected with the indicated plasmids and TRIM25 plasmid with and without poly(I:C) treatment. Anti-Flag immunoprecipitates were analyzed by immunoblot. **c** HEK293T cells were transfected with plasmids encoding TRIM25 WT and its mutants, then treated with poly(I:C) for 12 h. The p-IRF3 was then measured. **d** HEK293T WT or G3BP1 KO cells were transfected with the indicated vector and TRIM25 plasmids, the levels of p-IRF3 were analyzed by immunoblot. **e, f** WT or G3BP1 KO HEK293T cells were first transfected with TRIM25 WT or ΔPTFG, then treated with or without poly(I:C). The mRNA levels of IFNβ (**e**) and ISG56 (**f**) were

determined using qPCR. Poly(I:C) (−) vs. Poly(I:C) (+): *p* value: <0.001 (**e, f**), Vector (+) vs. mCh-TRIM25 (+): *p* value: <0.0001 (**e, f**), V2 (+) vs. G3BP1 KO (+): *p* value: <0.0001 (**e, f**), V2-mCh-TRIM25 (+) vs. G3BP1 KO-mCh-TRIM25 (+): *p* value: <0.0001 (**e, f**), G3BP1 KO-V2 (+) vs. G3BP1 KO-mCh-TRIM25 (+): *p* value: 2.96e−003 (**e**); 0.4607 (**f**). **g–j** HEK293T cells were transfected with TRIM25 WT, TRIM25 PTFG$^{AAAA}$ or ΔPTFG, then treated with poly(I:C) for 12 h. The mRNA levels of IFNα (**g**), IFNβ (**h**), IFNγ (**i**), and ISG56 (**j**) were determined using qPCR. Poly(I:C) (−) vs. Poly(I:C) (+): *p* value: 8.57e−004 (**g**); <0.001 (**h, j**); 1.02e−003 (**i**), Vector (+) vs. WT (+): *p* value: <0.0001 (**g–j**), WT (+) vs. PTFG$^{AAAA}$ (+): *p* value: <0.0001 (**g–j**), WT (+) vs. ΔPTFG (+): *p* value: <0.0001 (**g, h, j**); 2.97e−004 (**i**). All data are representative of at least three independent experiments (**a–j**). Mean ± s.d., statistical analysis was performed using one-way ANOVA (**c, e–j**).

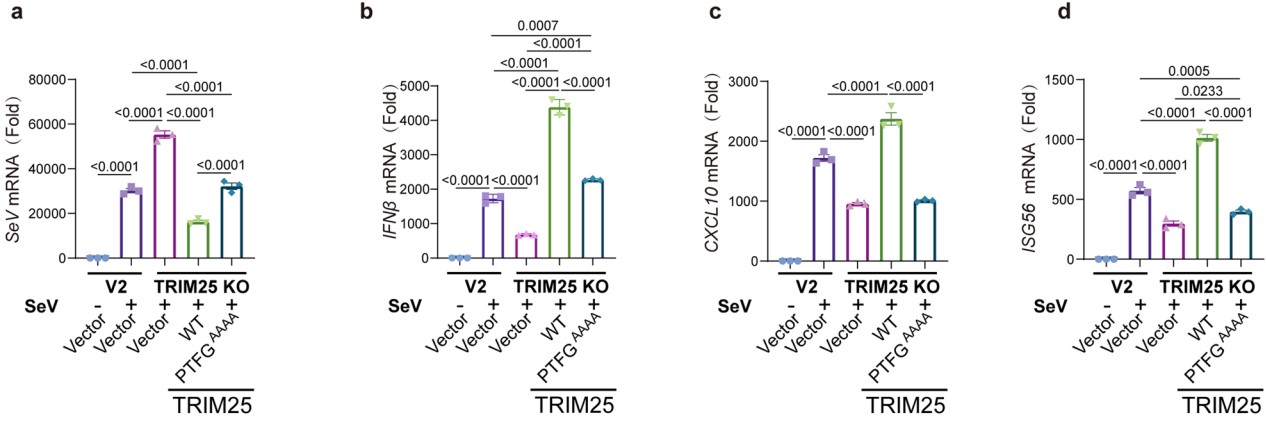

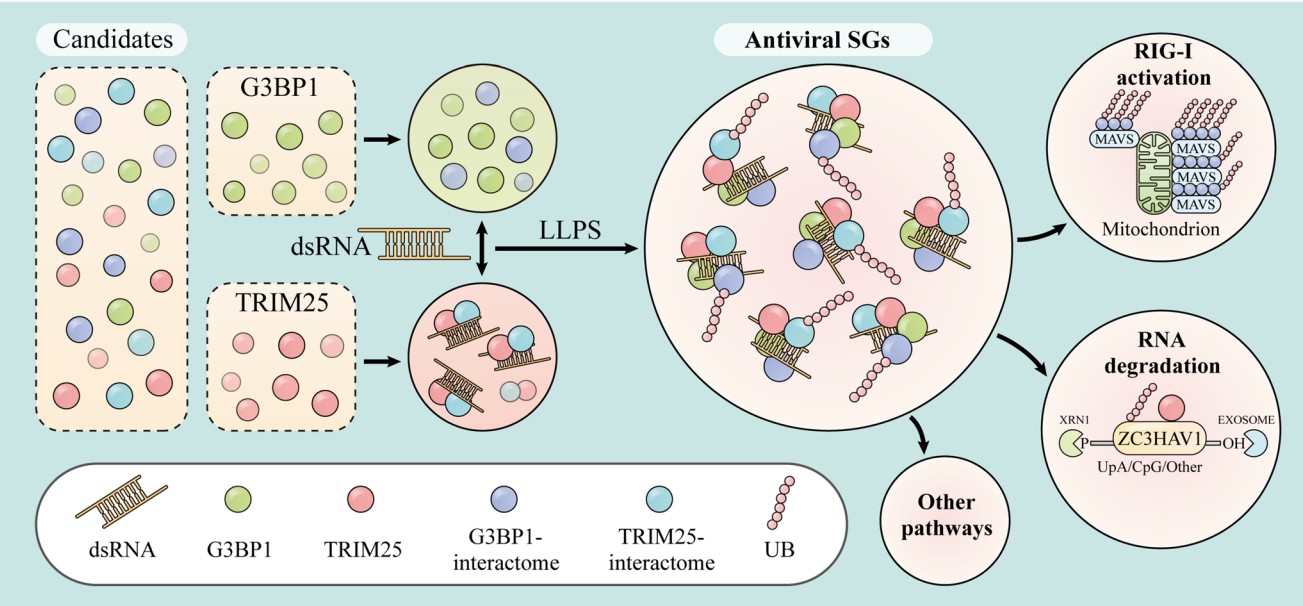

**Fig. 7 | Co-condensation of TRIM25 and G3BP1 inhibits SeV virus replication. a–d** WT (V2) or TRIM25 KO HEK293T cells were transfected with indicated plasmids, and then infected with SeV. **a** SeV mRNA level was determined 16 h post infection. The mRNA levels of host IFNβ (**b**), CXCL10 (**c**), and ISG56 (**d**) were determined 12 h post infection using qRT-PCR. SeV (−) vs. SeV (+): *p* value: <0.001 (**a–d**), V2 (+) vs. TRIM25 KO (+): *p* value: <0.001 (**a–d**), SeV (−) vs. SeV (+): *p* value: <0.001 (**a–d**), TRIM25 KO-Vector (+) vs. TRIM25 KO-TRIM25 WT (+): *p* value: <0.001

(**a–d**), TRIM25 KO-TRIM25 WT (+) vs. TRIM25 KO-TRIM25 ΔPTFG (+): *p* value: <0.001 (**a–d**). **e** A model showing how the co-condensation between G3BP1 and TRIM25 promotes the formation of antiviral SGs. This enhances TRIM25-mediated ubiquitination of many substrates, and could activate multiple antiviral pathways, such as the RIG-I pathway and ZC3HAV1-mediated RNA degradation. All data are representative of at least three independent experiments (**a–d**). Mean ±s.d., statistical analysis was performed using one-way ANOVA (**a–d**).

## RNA preparation

RNA oligonucleotides were purchased from Sangon BioTech either in single-stranded or double-stranded form. Sequences of the "sense" strands are (5′ to 3′): 14-mer, AUGGCUA GCUGGAG; 28-mer, AUGGCUA GCUGGAG CCACCCG CAGUUCG; 56-mer, AUGGCUA GCUGGAG CCACCCG CAGUUCG UACAAGG AUGACGA UGACAAG GGUACAA. DNA oligos had identical sequences, except that T was substituted for U[32].

## In vitro phase separation assay

TRIM25 and G3BP1 proteins were labeled by fluorescence dye Oregon-Green488 (Invitrogen, 2161802) and Cy3 (Lumiprobe, #41070), respectively. Purified proteins and fluorescence dyes were incubated with a molar ratio of 10:1. After reaction at 25 °C for 1 h, the excess fluorescence dye was removed by desalting column. The labeled proteins were concentrated to 10 mg/ml and stored in −80 °C.

The phase separation assays were performed at the room temperature, following an established protocol[78]. MBP-tagged TRIM25 (WT and ΔPTFG) and G3BP (WT, GDM and TBM) were mixed with 20 ng nucleic acids (dsRNA, ssRNA, dsDNA or ssDNA) in solution containing 20 mM Tris-HCl, pH 7.5, 150 mM NaCl, 8% v/v Ficoll. TEV protease was added at 0 min to start the reaction. The formation of droplets was observed using Olympus FV3000 confocal microscope with a 60x objective.

## Isothermal titration calorimetry (ITC)

ITC experiments were performed at 25 °C on a Microcal iTC200 instrument, similar to previous studies[25,79,80]. Prior to the titration, both proteins and peptides were dialyzed to the ITC buffer containing 25 mM HEPES (pH 7.5), 150 mM NaCl. The N protein peptide or TRIM25 peptide (700 μM) was titrated to G3BP1 NTF2 domain (50 μM) in 19 injections, with 2.0 μl each injection. Each experiment was performed at least three times. The data was analyzed using MicroCal PEAQ-ITC analysis Software.

## In vitro ubiquitination assays

The ubiquitination assays were performed as described previously[53,55]. In brief, the reaction mixture included 0.5 μM E1, 2.5 μM UbcH5b, 1 μM TRIM25, 8 μM Ub, 10 μM G3BP1 (WT, TBM, or GDM), 56-bp dsRNA, and 2 μM RIG-I-2CARD-HA or 2 μM ZC3HAV1-N-HA in buffer containing 50 mM HEPES, pH 7.5, 150 mM NaCl, 1 mM DTT, 5 mM ATP, and 5 mM MgCl$_2$. The reactions were carried out at 30 °C for 30 min, and terminated by adding the SDS loading buffer. The samples were then subjected to SDS-PAGE followed by immunoblotting using antibodies against Ub or HA.

## Statistical analysis

Each experiment was performed at least three times, and all data were presented as means ± SEM. Comparisons between groups were analyzed by unpaired two-tailed Student's $t$ test or one-way/two-way analysis of variance (ANOVA). Statistical analysis was performed using GraphPad Prism 8.

## Reporting summary

Further information on research design is available in the Nature Portfolio Reporting Summary linked to this article.

## Data availability

The mass spectrometry raw data were deposited to the ProteomeXchange Consortium with dataset identifier IPX0006706000 via the iProx partner repository. The data are accessible in iProX at the following link: https://www.iprox.cn/page/project.html?id=IPX0006706000. All data supporting the findings of this study are available within the article and its Supplementary files and from the corresponding author upon request. Source data are provided with this paper.

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

## Acknowledgements
This work was supported by National Key Research and Development Program of China (2022YFA1105200), Natural Science Foundation of China (NSFC, #92254302) and National Science Fund for Distinguished Young Scholars (#32125012). This work also instructed by the MGCJ organization.

## Author contributions
Z.S. and D.J. conceived the project. Z.S. designed and performed most cellular experiments with assistance from S.Z., X.Z. and J.W. S.Z. performed ITC, in vitro Phase Separation and ubiquitination assays, and L.Z. carried out viral infection experiment. D.D.B., P.Y. and L.Z. provided critical comments and suggestions. D.J., P.B. and F.Z. supervised the project. Z.S. and D.J. prepared the manuscript with input from every author.

## Competing interests
The authors declare no competing interests.
