## [Peer Review File · Nature Communications]

Reviewers' Comments:

Reviewer #1:

Remarks to the Author:

The stress granule subtype is an emerging concept in recent years. Different types of stress conditions with varying components, functions, etc. are formed as a result of various stress conditions. In this manuscript, the authors concentrated on virus-induced stress granules and did a proximal labeling experiment to identify the components and validate the active functions. Specifically, the authors found that TRIM25, a known driver of the ubiquitination-dependent antiviral innate immune response, increased most robustly among all proteins and was enriched upon poly(I:C) treatment, which is an analog of dsDNA to mimic virus infection. They further demonstrated the binding site between TRIM25 and G3BP1, and showed their interaction is essential for the recruitment of TRIM25 into stress granules upon poly(I:C) induction. Moreover, they showed how the co-condensation of TRIM25 and G3BP1 promoted ubiquitination activity and the RIG-1 signaling pathway. Stress granule as a platform for the assembly of the antiviral signaling pathway. Therefore, this discovery reveals a novel path for virus-activated immune response and is crucial to understanding how viruses initiate phase separation procedures and influence downstream pathways.

While the results are thorough, a few points should be addressed to further improve this work.

Major points:

1. In Figure 1c, the authors did comprehensive control experiments utilizing a number of different stress inducers to confirm that the recruitment of TRIM25 to the G3BP1 stress granule is specific to the poly(I:C). To make the comparison more thorough, a few typical stress conditions should be taken into account, such as heat, hypoxia, or osmotic stresses.
2. For Figure 1d, I suggest the author provide a more quantitative description of the stress granule and TRIM25 puncta. For example, it would be useful to know how much TRIM25 originates from G3BP1 independent puncta and how much co-localize with G3BP1.
3. In Figure 2d, the author demonstrated how 58-bp dsRNA might increase the LLPS of TRIM25. In order to demonstrate that the enhancement is caused by specific binding and not the crowding effect of RNA molecules, I suggest including a control experiment, using 58-bp dsRNA with scrambled or totally different sequences.
4. The authors showed TRIM25 interacts with G3BP1 through the PTFG motif. While the results of the truncation experiment are solid, the figures should be presented with more care. The G3BP1 condensate in Figure 2h and Figure 3e were depicted in quite distinct ways.
5. SARS-CoV-2 N protein disrupted the TRIM25-G3BP1 interaction, while H1N1 NS1 protein showed similar effect. Could the authors show more examples to demonstrate the specificity of the virus? Also, how the N protein competes with the TRIM25-G3BP1 interaction is an intriguing question. The authors might consider moving them to the main figures.
6. The authors demonstrated that the co-condensation between TRIM25 and G3BP1 promotes the ubiquitination activity of TRIM25, using western blot assay. Could they show other assays to show the ubiquitination activity is being carried out in condensates? For example, by immunofluorescence?

Reviewer #2:

Remarks to the Author:

In the manuscript Shang et al. presents a study to unravel the intricate aspects of stress granules and their involvement in the response to dsRNA viral infection. The authors identified TRIM25 as a specific marker protein involved in the formation and antiviral function of these stress granules. Adequate evidence is provided to demonstrate that TRIM25 undergoes Liquid-Liquid Phase Separation with the stress granule core, both in vitro and in cells. Additionally, the presence of

dsRNA facilitates the co-condensation of TRIM25 with stress granule components. Furthermore, the co-condensation of TRIM25 and stress granules plays a crucial role in regulating the dsRNA-responding antiviral signaling pathway by enriching numerous antiviral proteins. However, despite the promising findings, several major issues need to be addressed before it can be considered for publication. These issues should be thoroughly resolved to enhance the overall quality and scientific rigor of the manuscript.

Major points:

1. In Figure 1c, the authors reported that TRIM25 and G3BP1 formed colocalized puncta only when cells were infected with SeV or treated with poly(I:C). However, as TRIM25 was overexpressed in this experiment (whether G3BP1 was endogenous or overexpressed is not explicitly stated), it is essential to include control imaging in the experimental design to better interpret the results.
2. In Figure 1 d, the authors mentioned that poly(I:C) stimulation could induce two types of G3BP1-positive puncta: canonical stress granules (SG) and another type referred to as RLBs (non-canonical SG?). However, the manuscript lacks detailed evidence to depict the diameter difference between these two types of puncta. It is challenging for readers to discern the difference solely based on the presented immunofluorescent images. To strengthen the findings, it would be helpful to provide sufficient evidence, such as using a specific molecular marker for RLBs shown simultaneously.
3. In Figure 2b, the authors observed that recombinant TRIM25 could undergo spontaneous LLPS in vitro by adding TEV to MBP-tagged TRIM25 in the reaction. However, given that the manuscript mentions using His-tagged TRIM25 elsewhere, it is advisable to perform the experiment using His-tagged TRIM25, which would be more consistent. Furthermore, it would be more compelling to demonstrate the spontaneous droplet formation of TRIM25 without involving MBP peptide or TEV in the scenario. This approach would provide stronger evidence for the intrinsic LLPS behavior of TRIM25.
4. In Figure 2f, the authors reported that only dsRNA could promote the co-condensation between TRIM25 and G3BP1 after testing three other nucleic acids, including ssRNA, ssDNA, and dsDNA. However, detailed information about these other three nucleic acids, such as their length and base composition, was not provided in the manuscript. While it is mentioned that all nucleic acids used in the experiment were 20 ng, the molar concentration could vary significantly due to the unspecified length and base composition. Considering the importance of nucleic acid characteristics in explaining their interactions with proteins in vitro, it is essential to specify these details. Additionally, the use of fluorescence-labeled dsRNA as a ligand in the experiment and conducting experiments involving dsRNA incubation could help validate its beneficial effect during the co-condensation between TRIM25 and G3BP1.
5. In Figure 2 g-i, It is crucial to determine whether G3BP1 alone is capable of inducing spontaneous LLPS in vitro. Addressing this question is of great importance when interpreting the findings in panels g-i.
6. In Figure 4, the manuscript presents intriguing findings where both the overexpression methods and poly(I:C) stimulation show the same pattern of antiviral proteins (AVPs) enrichment. However, it remains unclear why these strategies exhibit similar patterns. It is essential to provide a plausible explanation for this observation. For instance, it should be clarified whether overexpressed TRIM25 readily condensates with AVPs in a quiescent condition, dependent on its interaction with G3BP1. Moreover, if the overexpression system does not reflect a normal physiological state, this should be addressed, and the implications for its use as a model for studying physical interactions need to be carefully considered and justified.
7. In Figure 4, the manuscript raises questions about why both the overexpression methods and poly(I:C) stimulation exhibit the same pattern of AVPs-enriched results. It is important to address whether overexpressed TRIM25 readily condenses with these AVPs in a quiescent condition, possibly dependent on its interaction with G3BP1. If the overexpression system does not accurately represent the normal physiological state, the rationale for its use in a regular physical study needs to be carefully evaluated and justified.
8. In Figure 6a, considering possible structural changes and the harshness of mutating all four alanine residues in the PTFG motif at once, it is advisable to conduct individual amino acid substitutions one at a time. This approach would allow the examination of how each PTFG-mutated form affects the ubiquitination of RIG-I and better understand its impact on TRIM25's normal RIG-I related function.
9. In Figure 6d, the authors reported that the deletion of G3BP1 abolishes TRIM25-induced p-IRF3

levels, based on a Western blot showing diminished p-IRF3 signal in the G3BP1 KO cell line. However, careful examination of the blot reveals that the basal p-IRF3 signal intensity in the two cell lines (wild-type and KO) is not relatively the same, considering the equivalent amount of IRF3 shown. Consequently, it is not appropriate to simply compare the signals under stimulation and draw conclusions about the impact of G3BP1 knockout on TRIM25-induced p-IRF3 levels. To strengthen the conclusion, it is necessary to provide direct evidence, such as qPCR analysis of IFNs, to illustrate the role of G3BP1 in TRIM25-induced antiviral signaling.

Minor points,

In Figure 3, these two panels appear to be cropped from a larger figure, and the signs are somewhat confusing.

In Figure 5a, the sentence describing the experiment sequence appears to be in the wrong order, and it should be revised accordingly.

Reviewer #3:

Remarks to the Author:

TRIM25 is an RNA-binding protein with E3 ubiquitin ligase activity involved in the positive regulation of innate immune response effectors such as RIG-I. Using a proximity-dependent biotinylation approach coupled to mass spectrometry analyses, the authors identify TRIM25 as a protein enriched in stress granules (SGs) induced by viral infection. They demonstrate TRIM25 ability to condense using in vitro liquid-liquid phase separation (LLPS) assays and show that interaction with G3BP1 via a motif identified in this study, the PTFG motif, promotes its ubiquitinating activity of several antiviral factors, including RIG-I, probably via its recruitment to SGs.

Although the colocalization of TRIM25 with RIG-I and G3BP1 in SGs and the importance of G3BP1 for TRIM25-mediated activation of RIG-I have already been reported by others, the authors here identify important domains in TRIM25 and propose the importance of its localization in SGs for the activation of antiviral proteins, a particularly interesting concept. This manuscript provides a wealth of data contributing resources on the interaction partners of G3BP1 and TRIM25, and a fine combination of state-of-the-art biochemical assays with in vitro experiments. However, key conclusions are drawn with TRIM25 mutants expressed in cells in which endogenous TRIM25 is still present and could affect the results. These findings need to be confirmed in reconstituted TRIM25 knockout cells. Furthermore, treatment with dsRNA is used as a marker of localization in SGs. However, as TRIM25 and G3BP1 are both able to bind dsRNA, some key in vitro ubiquitination activity experiments are lacking to confirm that the presence of dsRNA alone does not influence TRIM25 increased activity observed toward the targets. Finally, two mutants which lost the ability to bind G3BP1 are still capable to drive innate immune activation. This result weakens the conclusions on the antiviral stress granules (see comments below). More generally, although LLPS experiments are very interesting and could be informative, none of the conclusions drawn about droplet size or number are supported by quantifications. Finally, the title is misleading. TRIM25 is also found in arsenite-mediated SGs and its activity occurs also in absence of stress and viral infection.

Major comments:

- Figure 1: The authors dissect the G3BP1 interaction network in 293T cells treated or not with poly(I:C) and identify TRIM25 as the most highly differentially enriched candidate. TRIM25 colocalizes with G3BP1 in SeV- and poly(I:C)-induced SGs. But it forms distinct foci that remain associated with the periphery of SGs induced by other stresses such as arsenite. However, TRIM25 fluorescence signals overlap with that of G3BP. Co-localization analysis, quantifications should be provided. The number of repeats is unclear.
- - TRIM25 has previously been identified as an interaction partner of G3BP1 in SGs isolated from arsenite-treated U2OS cells (Jain et al., 2016) as well as by Markmiller et al. (2018) using the same cells and methodological approach as the authors here. Why was the dataset obtained (Figure S1d) compared with that from Marmor-Kollet et al. (2020)? In the latter study, 3 different baits were used to characterize SGs in sodium arsenite-treated U2OS cells and TRIM25 was not identified. This is neither mentioned in the text nor discussed.
- Figure 1d: quantification is missing. The authors make a strong point on the size difference between SGs and potential RNaseL bodies. Any other RNaseL body marker to substantiate this

hypothesis?

- What is the expression level of the endogenous TRIM25 protein in all the cells used in this study? Where does the endogenous protein localize under stress conditions? Is it comparable to the localization of the exogenous GFP-TRIM25 protein?

- Figure 2a: FRAP experiments in TRIM25-GFP-expressing cells show mobility of TRIM25 in TRIM25-positive foci (not droplets). This experiment was performed in presence of dsRNA but the statement is unclear in the text and in the legend.

- Figure 2b-d: In vitro LLPS analyses show a clear propensity for TRIM25 to phase separate in the absence of dsRNA. The summary graph provided in panel d, but also those in panels k and l, do not include the higher protein concentrations (25 and 50 μ M) and the absence of dsRNA (0 ng/ μ l). This does not correspond to the selected images. The dsRNA used is 58bp-long. How does the concentration chosen in vitro reflects transfection with poly(I:C) in cells?

- Figure 2e-f: The authors report a difference in TRIM25-G3BP1 droplet size in the presence of dsRNA over time. The conclusion should be supported by quantifications (and the number of replicates indicated) as shown in panel 2i. In addition, if the authors state in the main text, line 154, that the number of droplets increases in the presence of dsRNA. This is not quantified. All experiments are performed in the presence of a high concentration of molecular crowder. Can the protein condense in the absence of molecular crowder?

- Figure 2g-i: Mixture of TRIM25 and G3BP1 recombinant proteins. The experimental procedure is unclear both in the materials and methods section and in the legend. What is the concentration of protein used in the single sample versus the mixture? Are the protein concentrations adjusted in the mixture? The images show that droplet fusion occurs in the absence of dsRNA, but this is not reflected in the summary representations shown in panels k and l.

- Figure 3b: Do endogenous G3BP1 and TRIM25 interact? Provide missing quantifications and number of repeats. Is the interaction between G3BP1 and TRIM25 enhanced or modified when cells are treated with poly(I:C), considering that TRIM25 can bind dsRNA directly? Pull-down assays are important to substantiate this aspect. In addition, quantification of the interaction and the number of repeats is lacking.

- Figure 3c: Labeling of still images is barely visible. The authors could choose a more representative cell transfected with the TRIM25 mutant. This one looks rather unhealthy. Figure 3d provides images of untreated controls and GFP-TRIM25 WT. The authors mention line 210 "co-condensation frequency". This is a co-localization and quantification is missing.

- Figure 3e: the authors generate or LLPS assays and conclude that it is unable to increase G3BP1 droplet size. Quantification is lacking here. The experiment with the WT protein should be carried out in parallel. Why doesn't the G3BP1 protein alone form droplets as shown in figure 2h? What is the impact of dsRNA on the mutant?

- - Figure 4 provides data on the TRIM25 interactome compared with that of the mutant that cannot bind G3BP1. The enrichment analysis presented in panel 4d demonstrates the clear interaction of TRIM25 with antiviral proteins in the absence of stress in a G3BP1-dependent manner. This interaction also appears to be enhanced by poly(I:C) treatment. RIG-I/DHX58 does not appear in the curated list. What is the fold enrichment of RIG-I in both conditions? Is the degree of colocalization of antiviral factors and RIG-I correlated with their enrichment? The cells used for colocalization analyses (Figure 4e and S4d) express endogenous TRIM25. Since TRIM25 is capable of forming dimers, and a certain degree of colocalization of TRIM25 mutants has yet to be observed, this experiment must be carried out in the context of reconstituted knockout cells. The authors already have such a knockout cell line, as shown in Fig. 5. Can TRIM25 localize with antiviral proteins in a G3BP1 knockout context?

- Figure 5a confirms that SGs are sites in which ubiquitinated proteins accumulate, both in response to poly(I:C) and arsenite treatment, and that ubiquitination is mediated by TRIM25. This result contrasts with the fact that TRIM25 is not localized in the SGs of HeLa cells, as shown in Figure 1c, and also contradicts the title of the manuscript. Is the ubiquitin signal restored by expression of WT and mutant GFP-TRIM25 in knockout cells?

- Figures 5c-d show that the activity of TRIM25 is potentiated by G3BP1 and additionally by treatment with poly(I:C) in WT cells. However, to conclude about the importance of the localization in SGs, these experiments should be complemented by the use of GFP-TRIM25 Δ PTFG mutant. Furthermore, the in vitro ubiquitination assay (panel d) was performed in presence of dsRNA. What is the contribution of dsRNA to the enhanced TRIM25 activity in this assay, in presence and absence of G3BP1? This experiment will clarify whether the phenotype relies on TRIM25 activity, binding to dsRNA or to its localization in SGs. In the same line, experiments shown in panels f and

g were performed in the absence of dsRNA.

- Figure 5c-d: quantifications of the p-IRF3 levels and number of repeats is missing. The authors conclude in the main text, lines 374-375, that TRIM25 mutants Δ SPRY, 7KA and Δ CC are all impaired in IRF3 phosphorylation. However, the immunoblot shows similar levels of p-IRF3 for 7KA and Δ CC than WT, two mutants for which the interaction with G3BP1 is shown to be lost in Figure 3b. This observation undermines the general conclusion regarding the role of TRIM25 localization in SGs on the antiviral function of SGs.

- Figure 7a-e: Reconstitutes TRIM25 knockout cells must be used for this experiment. The presence of endogenous TRIM25 makes it difficult to conclude.

- Recent publication reports the ubiquitination of G3BP1 by TRIM25 (Yang Y, et al. 2023, *Biochim Biophys Acta Gene Regul Mech.* 10;1866(3):194954. doi: 10.1016/j.bbagr.2023.194954. PMID: 37302696). This new reference should be discussed.

Minor comments:

- Figure 1b: add the untreated panel.

- Figure 1e: label e is missing in the figure.

- Figure S1: The use of the terminology WT vs Poly(I:C) is not the best appropriated, rather untreated vs poly(I:C).

- Figure 2. Legend of panel 2a states that the data are presented are means \pm SEM at each time point. The panel only shows still images of the FRAP experiment, no recovery value is indicated. Naming the TRIM25-positive foci in cells "droplets" is also confusing.

- Text line 187: TRIM25 and G3BP1 undergo dsRNA-dependent condensation: this is not correct. dsRNA promotes condensation but is dispensable.

- Figure 7a-e, the legend states that TRIM25 PTFG-AAAA and Δ PTFG mutants were used. Only results for the mutant PTFG-AAAA are shown.

Reviewer #4:

Remarks to the Author:

Overview:

In this manuscript, "TRIM25 specifies antiviral stress granules", Shang et al. describe the co-condensation of TRIM25 and G3BP1 to stress granules in a dsRNA-dependent manner. Using a multidisciplinary approach of biochemistry, biophysics, and cell biology, the authors provide insights into how these proteins regulate the dynamics and formation of droplets in vitro and stress granules in vivo. A central conclusion of this manuscript is that through multiple distinct interactions, TRIM25 and G3BP1 promote stress granule formation to activate the RIG-I antiviral signalling pathway. While interesting, many of the results in this manuscript have been reported (Sanchez et al., 2018; Yang et al., 2022), which reduces the novelty of these findings. Another concern is that several experiments – for example, the ubiquitination assays – should have been performed more rigorously to justify the conclusions in the text (see points below). It is because of these reasons that I cannot, at this point, recommend this manuscript for publication, but I sincerely hope the authors find these comments helpful for further improving their manuscript.

Major points:

1. Line 134 & Fig. 1e, the authors should quantify the puncta/cell and display the data as a bar graph with errors.

2. Line 154 & Fig. S2c, the authors should quantify this difference (number & size) and display the data as a bar graph with errors.

3. Fig. 3c, the authors should quantify the difference in co-condensation frequency at the 8 h time point (number & size) and display the data as a bar graph with errors.

4. Fig. 3d-e, the bottom panel of this figure is not visible; thus, the data could not be assessed.

5. Fig. 5b-c, using this approach, you cannot determine the levels of substrate ubiquitination. The authors should first enrich ubiquitin (e.g., with tandem ubiquitin-binding entities -TUBEs) and then

blot for the substrates (IGF2BP2 and ZC3HAV1). This would allow for a direct readout of substrate ubiquitination levels.

6. Fig. 5d & Fig. S6a should have a negative control. The authors should spike the ITFG peptide into these reactions to determine if this interferes with TRIM25 substrate ubiquitination.

7. As RNA binding is also known to enhance TRIM25 activity (Sanchez et al. 2018), it would be interesting to compare RNA- vs. G3BP1-mediated TRIM25 activation in Fig. 5d.

8. In the method section for the in vitro ubiquitination assays, the authors report only using the E2 enzyme Ubc13 in their assays. However, Mms2/UBE2V2 should also be included in these assays since Ubc13/Mms2 forms a heterodimer to assemble Lys63 ubiquitin chains.

9. For Fig. 6a-b, see point 5 above.

10. Lines 373-376, the authors state, "Similarly, we also found that the E3 ligase activity (TRIM25 CS), RNA binding (Δ SPRY and 7KA), and dimerization (the coiled-coil mutant Δ CC) of TRIM25 were all required to fully phosphorylate IRF3 following poly(I:C) treatment." However, the levels of IRF3-p look the same as WT.

11. Line 376-377, the authors state, "Consistently, we also found that deletion of G3BP1 abolished TRIM25-induced p-IRF3 levels". However, IRF3-p levels are still increasing in a TRIM25-dependent manner in the G3BP1 KO background. The authors should modify their conclusion.

Minor points:

1. In the figure legends, the authors should indicate how many replicates were performed for each experiment.

2. In Fig. 2a legend, the authors state, "Data are presented as mean plus/minus standard error of the mean (s.e.m.) at each time point from three independent replicates." However, there is no data presented this way in the figure.

3. Line 110-111, the authors state, "Thus, poly(I:C) treatment stimulates the recruitment of TRIM25 into the SGs"; however, this is not shown until the following figure (Fig. 1c).

4. Fig.1e, the "e" is missing in the figure.

5. In Fig. 3c, the labels within the figure must be clearly visible (i.e., text in left panels).

6. In the WT vs. Δ PTFG interactome study, was this experiment performed in cells treated with poly(I:C)?

7. Fig. 7a-d, statistical analysis should be performed as in Fig. 6e.

Point-by-point responses:

We appreciate that the reviewers' constructive suggestions and the comment regarding the conceptual advance of the work. We have revised the manuscript extensively in response to the reviewers' concerns, and believe that this has greatly improved the quality of our manuscript. Our point-by-point responses are listed below:

REVIEWER COMMENTS

Reviewer #1 (Remarks to the Author):

The stress granule subtype is an emerging concept in recent years. Different types of stress conditions with varying components, functions, etc. are formed as a result of various stress conditions. In this manuscript, the authors concentrated on virus-induced stress granules and did a proximal labeling experiment to identify the components and validate the active functions. Specifically, the authors found that TRIM25, a known driver of the ubiquitination-dependent antiviral innate immune response, increased most robustly among all proteins and was enriched upon poly(I:C) treatment, which is an analog of dsDNA to mimic virus infection. They further demonstrated the binding site between TRIM25 and G3BP1, and showed their interaction is essential for the recruitment of TRIM25 into stress granules upon poly(I:C) induction. Moreover, they showed how the co-condensation of TRIM25 and G3BP1 promoted ubiquitination activity and the RIG-1 signaling pathway. Stress granule as a platform for the assembly of the antiviral signaling pathway. Therefore, this discovery reveals a novel path for virus-activated immune response and is crucial to understanding how viruses initiate phase separation procedures and influence downstream pathways.

While the results are thorough, a few points should be addressed to further improve this work.

Major points:

1. In Figure 1c, the authors did comprehensive control experiments utilizing a number of different stress inducers to confirm that the recruitment of TRIM25 to the G3BP1 stress granule is specific to the poly(I:C). To make the comparison more thorough, a few typical stress conditions should be taken into account, such as heat, hypoxia, or osmotic stresses.

Response: We appreciate the reviewer's thoughtful suggestions. To address this, we added osmotic and heat shock stress to our system, and found that SGs did not show strong colocalization with TRIM25 foci upon either type of stress (new Fig. 1c).

2. For Figure 1d, I suggest the author provide a more quantitative description of the stress granule and TRIM25 puncta. For example, it would be useful to know how much TRIM25 originates from G3BP1 independent puncta and how much co-localize with G3BP1.

Response: We appreciate this suggestion and have defined SGs and RLBs by the radius of the foci (new Fig. 1g) and stained with specific protein markers (new Fig. S1h). We found that TRIM25 strongly co-localized with RLBS and SGs, but the co-localization with SGs was stronger than with RLBs (new Fig. 1f).

3. In Figure 2d, the author demonstrated how 58-bp dsRNA might increase the LLPS of TRIM25. In order to demonstrate that the enhancement is caused by specific binding and not the crowding effect of RNA molecules, I suggest including a control experiment, using 58-bp dsRNA with scrambled or totally different sequences.

Response: We extend our gratitude for your insightful suggestion. To demonstrate that the enhancement is caused by specific binding and not the crowding effect of RNA molecules, we included 56bp-dsRNA^{TATA}, which is composed of multiple TA repeats, as a comparison. We found that both 56bp-dsRNA and 56bp-dsRNA^{TATA} effectively mediated the formation of TRIM25 droplets (new Fig. S2f). Interestingly, TRIM25 formed smaller droplets in the presence of 56bp-dsRNA (new Fig. S2g), suggesting that the LLPS properties of TRIM25 are regulated by the sequence features of dsRNA.

4. The authors showed TRIM25 interacts with G3BP1 through the PTFG motif. While the results of the truncation experiment are solid, the figures should be presented with more care. The G3BP1 condensate in Figure 2h and Figure 3e were depicted in quite distinct ways.

Response: Thank you for your valuable comments and suggestions on our manuscript. Following the suggestion, we have presented the original Fig. 2h along with Fig. 3e as new Fig. 3g-Fig. 3j with unified descriptions.

5. SARS-CoV-2 N protein disrupted the TRIM25-G3BP1 interaction, while H1N1 NS1 protein showed similar effect. Could the authors show more examples to demonstrate the specificity of the virus? Also, how the N protein competes with the TRIM25-G3BP1 interaction is an intriguing question. The authors might consider moving them to the main figures.

Response: Thank you for these suggestions. Multiple viral proteins bear the Φ xFG sequence motif and bind G3BP, including the non-structural protein 3 (nsP3) of the Old-World alphavirus Semliki Forest virus, and ICP8 of herpes simplex virus (PMID: 25658430; PMID: 27383630). Since characterizing of these interactions is not the main focus of our current study, we presented them in Fig. S3.

6. The authors demonstrated that the co-condensation between TRIM25 and G3BP1 promotes the ubiquitination activity of TRIM25, using western blot assay. Could they show other assays to show the ubiquitination activity is being carried out in condensates? For example, by immunofluorescence?

Response: In response to this suggestion, we performed immunofluorescence assays using antibodies against G3BP1 or K63-polyUb, and found that G3BP1 and K63-polyUb displayed strong co-localization in WT cells upon poly(I:C) stimulation (new Fig. S6a). Furthermore, deletion of TRIM25 dramatically diminished the co-localization (new Fig. S6a). Importantly, the diminished co-localization between G3BP1 and K63-polyUb could be rescued by the re-expression of TRIM25 WT, but not by TRIM25 Δ PTFG, emphasizing the importance of TRIM25-G3BP1 interaction in regulating the ubiquitination of SG proteins (new Fig. 5a).

Reviewer #2 (Remarks to the Author):

In the manuscript Shang et al. presents a study to unravel the intricate aspects of stress granules and their involvement in the response to dsRNA viral infection. The authors identified TRIM25 as a specific marker protein involved in the formation and antiviral function of these stress granules. Adequate evidence is provided to demonstrate that TRIM25 undergoes Liquid-Liquid Phase Separation with the stress granule core, both in vitro and in cells. Additionally, the presence of dsRNA facilitates the co-condensation of TRIM25 with stress granule components. Furthermore, the co-condensation of TRIM25 and stress granules plays a crucial role in regulating the dsRNA-responding antiviral signaling pathway by enriching numerous antiviral proteins.

However, despite the promising findings, several major issues need to be addressed before it can be considered for publication. These issues should be thoroughly resolved to enhance the overall quality and scientific rigor of the manuscript.

Major points:

1. In Figure 1c, the authors reported that TRIM25 and G3BP1 formed colocalized puncta only when cells were infected with SeV or treated with poly(I:C). However, as TRIM25 was overexpressed in this experiment (whether G3BP1 was endogenous or overexpressed is not explicitly stated), it is essential to include control imaging in the experimental design to better interpret the results.

Response: We thank you for your suggestion and have added control imaging to our experiments (new Fig. 1c). TRIM25 foci could be formed alone when the protein was over-expressed. G3BP1 was endogenous in this experiment, and we have updated our figure legend, as suggested.

2. In Figure 1 d, the authors mentioned that poly(I:C) stimulation could induce two types of G3BP1-positive puncta: canonical stress granules (SG) and another type referred to as RLBs (non-canonical SG?). However, the manuscript lacks detailed evidence to depict the diameter difference between these two types of puncta. It is challenging for readers to discern the difference solely based on the presented immunofluorescent images. To strengthen the findings, it would be helpful to provide sufficient evidence, such as using a specific molecular marker for RLBs shown simultaneously.

Response: We appreciate the excellent suggestion! As indicated above in response to reviewer 1, we defined SGs and RLBs by the radius of the G3BP1 foci (new Fig. 1g) and stained with specific markers (new Fig. S1h). The large puncta represented canonical SGs, and the small ones corresponded to the RNase L-dependent bodies (RLBs) identified in previous studies (PMID: 31896577). We confirmed the identity of the small puncta as RLBs by immunostaining with PABPC1 or TIA, protein makers used to distinguish RLBs from SGs. As expected for RLBs, these puncta co-stained with PABPC1, but not TIA (new Fig. S1h).

3. In Figure 2b, the authors observed that recombinant TRIM25 could undergo spontaneous LLPS in vitro by adding TEV to MBP-tagged TRIM25 in the reaction. However, given that the manuscript mentions using His-tagged TRIM25 elsewhere, it is advisable to perform the experiment using His-tagged TRIM25, which would be more consistent. Furthermore, it would be more compelling to demonstrate the spontaneous droplet formation of TRIM25 without involving MBP peptide or TEV in the scenario. This approach would provide stronger evidence for the intrinsic LLPS behavior of TRIM25.

Response: We appreciate your concern about the potential impact of the cleaved MBP tag and the TEV protease on TRIM25 LLPS. We performed His-tagged TRIM25 *in vitro* LLPS experiments

and found that it could also form TRIM25 droplets, similar to MBP-TRIM25 (Graphic 1). Furthermore, similar conditions were used in multiple published papers to study *in vitro* LLPS (such as PMID: 36738734; PMID: 34678144).

Graphic 1 His-tagged TRIM25 forms LLPS *in vitro*. Fluorescence microscopy images of 30 μM Alexa Fluor 488-labelled His-tagged TRIM25, mixed in a solution containing 20 mM Tris-HCl, pH 7.5, 150 mM NaCl and 8% v/v Ficoll in a solution containing 20 mM Tris-HCl, pH 7.5, 150 mM NaCl and 8% v/v Ficoll. Scale bar: 10 μm

4. In Figure 2f, the authors reported that only dsRNA could promote the co-condensation between TRIM25 and G3BP1 after testing three other nucleic acids, including ssRNA, ssDNA, and dsDNA. However, detailed information about these other three nucleic acids, such as their length and base composition, was not provided in the manuscript. While it is mentioned that all nucleic acids used in the experiment were 20 ng, the molar concentration could vary significantly due to the unspecified length and base composition. Considering the importance of nucleic acid characteristics in explaining their interactions with proteins *in vitro*, it is essential to specify these details. Additionally, the use of fluorescence-labeled dsRNA as a ligand in the experiment and conducting experiments involving dsRNA incubation could help validate its beneficial effect during the co-condensation between TRIM25 and G3BP1.

Response: Thank you for your insightful comments. We have added information of the four nucleic acids in the revised manuscript (line 696-702). Experiments were performed using same mass of nucleic acids (20ng). We also provided experimental details in the updated figure legends (line 812-813).

Additionally, we performed LLPS experiments with fluorescence-labeled dsRNA and found that co-condensation was formed between TRIM25, G3BP1 and dsRNA (new Fig. S2h).

5. In Figure 2 g-i, It is crucial to determine whether G3BP1 alone is capable of inducing spontaneous LLPS *in vitro*. Addressing this question is of great importance when interpreting the findings in panels g-i.

Response: Thanks a lot for your suggestions. Our new Fig. 2j shows that G3BP1 alone underwent spontaneous LLPS. The addition of TRIM25 reduced the protein and RNA concentration required for the formation of G3BP1 droplets (new Fig. 2j-l).

6. In Figure 4, the manuscript presents intriguing findings where both the overexpression methods and poly(I:C) stimulation show the same pattern of antiviral proteins (AVPs) enrichment. However, it remains unclear why these strategies exhibit similar patterns. It is essential to provide a plausible explanation for this observation. For instance, it should be clarified whether overexpressed TRIM25 readily condensates with AVPs in a quiescent condition, dependent on its interaction with G3BP1. Moreover, if the overexpression system does not reflect a normal physiological state, this should be

addressed, and the implications for its use as a model for studying physical interactions need to be carefully considered and justified.

Response: Thank you very much for this insightful suggestion. We suspect that TRIM25 and G3BP1, on its own, are able to associate with certain antiviral proteins (AVPs), and that the association is further enhanced due to TRIM25-G3BP1 interaction or upon poly(I:C) stimulation. Consistent with our hypothesis, we found that IGF2BP2 and ZC3HAV1 displayed strong co-localization with TRIM25 droplets even in G3BP1 dKO HeLa cells (new Fig. S4e). Similarly, RIG-I and ZCCHC3 showed pronounced co-localization with G3BP1, but not with TRIM25 Δ PTFG (new Fig. 4e).

7. In Figure 4, the manuscript raises questions about why both the overexpression methods and poly(I:C) stimulation exhibit the same pattern of AVPs-enriched results. It is important to address whether overexpressed TRIM25 readily condenses with these AVPs in a quiescent condition, possibly dependent on its interaction with G3BP1. If the overexpression system does not accurately represent the normal physiological state, the rationale for its use in a regular physical study needs to be carefully evaluated and justified.

Response: Thank you for your question. We suspect that TRIM25 and G3BP1, on its own, are able to associate with certain antiviral proteins (AVPs), and that the association is further enhanced due to TRIM25-G3BP1 interaction or upon poly(I:C) stimulation. Consistent with our hypothesis, we found that IGF2BP2 and ZC3HAV1 displayed strong co-localization with TRIM25 droplets even in G3BP1 dKO HeLa cells (new Fig. S4e). Similarly, RIG-I and ZCCHC3 showed pronounced co-localization with G3BP1, but not with TRIM25 Δ PTFG (new Fig. 4e).

8. In Figure 6a, considering possible structural changes and the harshness of mutating all four alanine residues in the PTFG motif at once, it is advisable to conduct individual amino acid substitutions one at a time. This approach would allow the examination of how each PTFG-mutated form affects the ubiquitination of RIG-I and better understand its impact on TRIM25's normal RIG-I related function.

Response: Following your suggestions, we performed single or double alanine substitution of the four amino acids in the PTFG motif (new Fig. S3d). All mutants abolished the interaction with G3BP1, as well as the ubiquitination of RIG-I, highlighting the importance of the integrity of the PTFG sequence (new Fig. S3d).

9. In Figure 6d, the authors reported that the deletion of G3BP1 abolishes TRIM25-induced p-IRF3 levels, based on a Western blot showing diminished p-IRF3 signal in the G3BP1 KO cell line. However, careful examination of the blot reveals that the basal p-IRF3 signal intensity in the two cell lines (wild-type and KO) is not relatively the same, considering the equivalent amount of IRF3 shown. Consequently, it is not appropriate to simply compare the signals under stimulation and draw conclusions about the impact of G3BP1 knockout on TRIM25-induced p-IRF3 levels. To strengthen the conclusion, it is necessary to provide direct evidence, such as qPCR analysis of IFNs, to illustrate the role of G3BP1 in TRIM25-induced antiviral signaling.

Response: The reviewer has made good points here, and we have updated our conclusions accordingly (Line 444-451). In addition, we performed quantitative RT-PCR (qPCR) analyses of

IFN β and ISG56 as suggested, and found that deletion of G3BP1 also decreased poly(I:C)-induced gene expression of IFN β and ISG56 (new Fig. 6e-f).

Minor points,

In Figure 3, these two panels appear to be cropped from a larger figure, and the signs are somewhat confusing.

Response: Thanks for your suggestion. Fig.3 has been rearranged to make it clearer.

In Figure 5a, the sentence describing the experiment sequence appears to be in the wrong order, and it should be revised accordingly.

Response: We apologize for the mistake. We have corrected it.

Reviewer #3 (Remarks to the Author):

TRIM25 is an RNA-binding protein with E3 ubiquitin ligase activity involved in the positive regulation of innate immune response effectors such as RIG-I. Using a proximity-dependent biotinylation approach coupled to mass spectrometry analyses, the authors identify TRIM25 as a protein enriched in stress granules (SGs) induced by viral infection. They demonstrate TRIM25 ability to condense using in vitro liquid-liquid phase separation (LLPS) assays and show that interaction with G3BP1 via a motif identified in this study, the PTFG motif, promotes its ubiquitinating activity of several antiviral factors, including RIG-I, probably via its recruitment to SGs.

Although the colocalization of TRIM25 with RIG-I and G3BP1 in SGs and the importance of G3BP1 for TRIM25-mediated activation of RIG-I have already been reported by others, the authors here identify important domains in TRIM25 and propose the importance of its localization in SGs for the activation of antiviral proteins, a particularly interesting concept. This manuscript provides a wealth of data contributing resources on the interaction partners of G3BP1 and TRIM25, and a fine combination of state-of-the-art biochemical assays with in vitro experiments. However, key conclusions are drawn with TRIM25 mutants expressed in cells in which endogenous TRIM25 is still present and could affect the results. These findings need to be confirmed in reconstituted TRIM25 knockout cells. Furthermore, treatment with dsRNA is used as a marker of localization in SGs. However, as TRIM25 and G3BP1 are both able to bind dsRNA, some key in vitro ubiquitination activity experiments are lacking to confirm that the presence of dsRNA alone does not influence TRIM25 increased activity observed toward the targets. Finally, two mutants which lost the ability to bind G3BP1 are still capable to drive innate immune activation. This result weakens the conclusions on the antiviral stress granules (see comments below). More generally, although LLPS experiments are very interesting and could be informative, none of the conclusions drawn about droplet size or number are supported by quantifications. Finally, the title is misleading. TRIM25 is also found in arsenite-mediated SGs and its activity occurs also in absence of stress and viral infection.

Response: Following your suggestions, we have changed the title of our manuscript to “TRIM25 predominately associates with anti-viral stress granules”. We also changed other parts of the manuscript accordingly.

Major comments:

- Figure 1: The authors dissect the G3BP1 interaction network in 293T cells treated or not with poly(I:C) and identify TRIM25 as the most highly differentially enriched candidate. TRIM25 colocalizes with G3BP1 in SeV- and poly(I:C)-induced SGs. But it forms distinct foci that remain associated with the periphery of SGs induced by other stresses such as arsenite. However, TRIM25 fluorescence signals overlap with that of G3BP. Co-localization analysis, quantifications should be provided. The number of repeats is unclear.

Response: Following your suggestions, we provided co-localization analysis (new Fig. 1c) and quantification (new Fig. 1d). All experiments are performed at least in triplicate, as indicated in the figure legend.

- - TRIM25 has previously been identified as an interaction partner of G3BP1 in SGs isolated from arsenite-treated U2OS cells (Jain et al., 2016) as well as by Markmiller et al. (2018) using the same cells and methodological approach as the authors here. Why was the dataset obtained (Figure S1d) compared with that from Marmor-Kollet et al. (2020)? In the latter study, 3 different baits were used to characterize SGs in sodium arsenite-treated U2OS cells and TRIM25 was not identified. This is neither mentioned in the text nor discussed.

Response: We appreciate your thoughtful comments. In the Marmor-Kollet et al. (2020) paper, the authors combined their own data and previously published data from Jain et al., (2016) and Markmiller et al. (2018), and formed a more comprehensive dataset (called “Combined SG proteome” in their paper). We thus compare our dataset with the combined dataset.

- Figure 1d: quantification is missing. The authors make a strong point on the size difference between SGs and potential RNaseL bodies. Any other RNaseL body marker to substantiate this hypothesis?

Response: We appreciate your valuable suggestion! As described above in response to reviewer 1 and 2, we defined SGs and RLBs using the radius of the G3BP1 foci and some specific markers. The large puncta represented canonical SGs, and the small ones corresponded to the RNase L-dependent bodies (RLBs) identified in previous studies (Fig. 1g). We also quantified the Pearson correlation coefficient of TRIM25 co-localization with SGs and RLBs. We found that TRIM25 strongly co-localized with RLBs and SGs, but the co-localization with SGs was stronger than with RLBs (new Fig. 1f). Following your suggestion, we confirmed the identity of the small puncta as RLBs by immunostaining with PABPC1 or TIA, protein markers used to distinguish RLBs from SGs. As expected for RLBs, these puncta co-stained with PABPC1, but not TIA (new Fig. S1h).

- What is the expression level of the endogenous TRIM25 protein in all the cells used in this study? Where does the endogenous protein localize under stress conditions? Is it comparable to the localization of the exogenous GFP-TRIM25 protein?

Response: Thank you for your question. TRIM25 is an ISG gene, and its expression is regulated by interferon stimulation. Indeed, we found that the TRIM25 protein level increased slightly over 2 folds upon poly(I:C) treatment (new Fig. S1f). In contrast, TRIM25 was enriched ~130 times in

SGs in the same condition (Fig. 1b). We also examined the localization of endogenous TRIM25. Similar to exogenous TRIM25, endogenous TRIM25 is also localized to SGs upon poly(I:C) treatment (new Fig. 1e).

- Figure 2a: FRAP experiments in TRIM25-GFP-expressing cells show mobility of TRIM25 in TRIM25-positive foci (not droplets). This experiment was performed in presence of dsRNA but the statement is unclear in the text and in the legend.

Response: We apologize for not explaining this point clearly in our original submission. Corresponding terminology and descriptions have been revised and added in the text and figure legends (Line 800-801).

- Figure 2b-d: In vitro LLPS analyses show a clear propensity for TRIM25 to phase separate in the absence of dsRNA. The summary graph provided in panel d, but also those in panels k and l, do not include the higher protein concentrations (25 and 50 μ M) and the absence of dsRNA (0 ng/ μ l). This does not correspond to the selected images. The dsRNA used is 58bp-long. How does the concentration chosen in vitro reflects transfection with poly(I:C) in cells?

Response: Thank you for pointing out the problem, we redesigned and performed the experiments. And the corresponding high-concentration protein phase diagram has been modified (new Fig. 2c; Fig. 2j-k). The concentration of dsRNA that we used was consistent with previous studies (PMID: 30342007). To confirm that co-condensation of dsRNA with TRIM25 and G3BP1 also occurs *in vivo*, we transfected cy5-tagged dsRNA to cells co-expressing mCherry-TRIM25 and GFP-G3BP1, and found that cy5-dsRNA co-localized with TRIM25 and G3BP1 in the cells (new Fig. S2i).

- Figure 2e-f: The authors report a difference in TRIM25-G3BP1 droplet size in the presence of dsRNA over time. The conclusion should be supported by quantifications (and the number of replicates indicated) as shown in panel 2i. In addition, if the authors state in the main text, line 154, that the number of droplets increases in the presence of dsRNA. This is not quantified. All experiments are performed in the presence of a high concentration of molecular crowder. Can the protein condense in the absence of molecular crowder?

Response: Following your suggestion, we have quantified the differences in TRIM25-G3BP1 droplet size and number over time in the presence of dsRNA (new Fig. S2e-h). Also, we used 8% v/v Ficoll as the molecular crowder. Without the crowder, we found that TRIM25 did not condense.

- Figure 2g-i: Mixture of TRIM25 and G3BP1 recombinant proteins. The experimental procedure is unclear both in the materials and methods section and in the legend. What is the concentration of protein used in the single sample versus the mixture? Are the protein concentrations adjusted in the mixture? The images show that droplet fusion occurs in the absence of dsRNA, but this is not reflected in the summary representations shown in panels k and l.

Response: Thank you for your constructive comments and questions. We apologize for not making it clear in the original submission. In this experiment, the concentration of both TRIM25 and G3BP1 was 50 μ M, and the protein concentration was reduced to 25 μ M in the mixture due to the dilution effect. Total dsRNA was included in this experiment. We have polished the text and figure legends accordingly (Line 258-262; Line 265-270; Line 847-849; Line 854-856).

- Figure 3b: Do endogenous G3BP1 and TRIM25 interact? Provide missing quantifications and number of repeats. Is the interaction between G3BP1 and TRIM25 enhanced or modified when cells are treated with poly(I:C), considering that TRIM25 can bind dsRNA directly? Pull-down assays are important to substantiate this aspect. In addition, quantification of the interaction and the number of repeats is lacking.

Response: We appreciate the reviewer's thoughtful suggestions. Using an antibody against TRIM25, we were able to detect the interaction between endogenous TRIM25 and G3BP1 (new Fig. S3a). Interestingly, poly(I:C) treatment did not further enhance the interaction (new Fig. S3b). All experiments are performed at least in triplicate.

- Figure 3c: Labeling of still images is barely visible. The authors could choose a more representative cell transfected with the TRIM25 mutant. This one looks rather unhealthy. Figure 3d provides images of untreated controls and GFP-TRIM25 WT. The authors mention line 210 "co-condensation frequency". This is a co-localization and quantification is missing.

Response: A new image is provided in the revised manuscript (new Fig. 3d). Furthermore, we quantified the shortest distance between TRIM25 WT and Δ PTFG to G3BP1 upon poly(I:C) or sodium arsenite stimulation (Graphic 2). Unlike TRIM25 WT (Fig. 1c), TRIM25 Δ PTFG did not co-localize with G3BP1 upon poly(I:C) stimulation (new Fig. 3e).

Graphic 2 Co-localization of TRIM25 and G3BP1 foci upon poly(I:C) or sodium arsenite treatment. HeLa cells expressing GFP-TRIM25 WT or Δ PTFG were treated with poly(I:C) or sodium arsenite, and then stained with an antibody against G3BP1. The shortest distance between TRIM25 foci and SGs was used for quantification.

- Figure 3e: the authors generate or LLPS assays and conclude that it is unable to increase G3BP1 droplet size. Quantification is lacking here. The experiment with the WT protein should be carried out in parallel. Why doesn't the G3BP1 protein alone form droplets as shown in figure 2h? What is the impact of dsRNA on the mutant?

Response: Thank you for your suggestions and questions. In the revised manuscript, we have quantified TRIM25 and G3BP1 droplet areas (new Fig. 3h; new Fig. 3j). We also put together experimental results with purified TRIM25 WT, Δ PTFG protein mixed with G3BP1 for comparison (new Fig. 3g-j). We believe that the difference in G3BP1 droplet size in the original manuscript is

due to image selection, and we have provided new representative image (new Fig. 3i). Finally, the effect of dsRNA on TRIM25 WT is essentially the same as that of TRIM25 Δ PTFG (new Fig. S2a; Graphic 3).

Graphic 3 TRIM25 WT and Δ PTFG LLPS are promoted by dsRNA to almost the same extent. MBP-TRIM25 WT or Δ PTFG (50 μ M) and 56bp-dsRNA (10 ng/ μ l) were mixed, and TEV was added at 0 min to cleave the MBP tag. Scale bar: 20 μ m. shows representative images of three independent experiments. Data are expressed as mean \pm s.e.m. Statistical significance was determined using the ordinary t-test.

- - Figure 4 provides data on the TRIM25 interactome compared with that of the mutant that cannot bind G3BP1. The enrichment analysis presented in panel 4d demonstrates the clear interaction of TRIM25 with antiviral proteins in the absence of stress in a G3BP1-dependent manner. This interaction also appears to be enhanced by poly(I:C) treatment. RIG-I/DHX58 does not appear in the curated list. What is the fold enrichment of RIG-I in both conditions? Is the degree of colocalization of antiviral factors and RIG-I correlated with their enrichment? The cells used for colocalization analyses (Figure 4e and S4d) express endogenous TRIM25. Since TRIM25 is capable of forming dimers, and a certain degree of colocalization of TRIM25 mutants has yet to be observed, this experiment must be carried out in the context of reconstituted knockout cells. The authors already have such a knockout cell line, as shown in Fig. 5. Can TRIM25 localize with antiviral proteins in a G3BP1 knockout context?

Response: We extend our gratitude for your insightful suggestions. Following your suggestions, we repeated the previous experiments in TRIM25 KO cells (new Fig. 4e). Our conclusions remain largely unchanged. RIG-I, ZC3HAV1, ZCCHC3, and IGF2BP2 displayed strong co-localization with both G3BP1 and TRIM25 upon poly(I:C) treatment, confirming our mass spectrometric results (new Fig. 4e). In cells expressing GFP-TRIM25 Δ PTFG, RIG-I and ZCCHC3 showed pronounced co-localization with G3BP1, but not with TRIM25 Δ PTFG (new Fig. 4e). IGF2BP2 showed significant co-localization with both G3BP1 and TRIM25 Δ PTFG droplets, consistent with its association with both proteins (new Fig.4e). Unlike them, ZC3HAV1 co-localized with TRIM25 Δ PTFG droplets, but not with poly(I:C)-induced SGs, likely reflecting a tight interaction between ZC3HAV1 and TRIM25 (new Fig. 4e).

We believe that the degree of co-localization of antiviral factors should correlate with the degree of enrichment in the proximity labeling assay. For example, ZCCHC3 is a known client protein of SGs and displays the largest difference in the degree of enrichment in the TRIM25 interactome (WT vs. Δ PTFG) (Fig. 4d). On the other hand, ZC3HAV1, a known client protein of TRIM25, displays the smallest difference (Fig. 4d).

Several published papers (PMID: 20403326; PMID: 31006531; PMID: 30193849) suggest that the interaction between RIG-I and TRIM25 may not be direct. Indeed, we did not identify RIG-I in the TRIM25 interactomes. Furthermore, we found the cellular localization of RIG-I depended on G3BP1, but not TRIM25 (Fig. 4e).

- Figure 5a confirms that SGs are sites in which ubiquitinated proteins accumulate, both in response to poly(I:C) and arsenite treatment, and that ubiquitination is mediated by TRIM25. This result contrasts with the fact that TRIM25 is not localized in the SGs of HeLa cells, as shown in Figure 1c, and also contradicts the title of the manuscript. Is the ubiquitin signal restored by expression of WT and mutant GFP-TRIM25 in knockout cells?

Response: We appreciate this suggestion! We found that the association between TRIM25 and G3BP1 is mainly driven by RNA viruses or dsRNA, and to a much less degree by arsenite treatment (Fig. 1c). Thus, we have changed the title of our manuscript to “TRIM25 predominately associates with anti-viral stress granules”.

Following your suggestions, we performed rescue experiment by expressing TRIM25 WT and mutant in TRIM25 knockout cells. We found that TRIM25 WT restored ubiquitin signaling in SGs, whereas GFP-TRIM25 Δ PTFG did not (new Fig. 5a).

- Figures 5c-d show that the activity of TRIM25 is potentiated by G3BP1 and additionally by treatment with poly(I:C) in WT cells. However, to conclude about the importance of the localization in SGs, these experiments should be complemented by the use of GFP-TRIM25 Δ PTFG mutant. Furthermore, the *in vitro* ubiquitination assay (panel d) was performed in presence of dsRNA. What is the contribution of dsRNA to the enhanced TRIM25 activity in this assay, in presence and absence of G3BP1? This experiment will clarify whether the phenotype relies on TRIM25 activity, binding to dsRNA or to its localization in SGs. In the same line, experiments shown in panels f and g were performed in the absence of dsRNA.

Response: Following your suggestions, we performed two sets of experiments and examined the effect of TRIM25 WT and Δ PTFG on substrate ubiquitination. First, we found that deletion of TRIM25 markedly diminished the ubiquitination level of endogenous IGF2BP2 in HEK293T cells treated with poly(I:C) (new Fig. 5b). Critically, the decreased ubiquitination level of IGF2BP2 could be rescued by re-expressing of TRIM25 WT, but not by that of TRIM25 Δ PTFG or CS mutants (Fig. 5b). Second, we also found that Poly(I:C) treatment significantly enhanced ubiquitination of ZC3HAV1 mediated by TRIM25 WT, but not by TRIM25 Δ PTFG (Fig.5c).

Furthermore, we also performed *in vitro* ubiquitination experiments in the presence or absence of G3BP1. G3BP1 enhanced the ubiquitination activity of TRIM25 in the presence of dsRNA (new Fig. 5d-e; new Fig. S6d). All *in vitro* ubiquitination experiments were performed in the presence of dsRNA, and we have clearly stated the experiment condition in the revised manuscript.

- Figure 5c-d: quantifications of the p-IRF3 levels and number of repeats are missing. The authors conclude in the main text, lines 374-375, that TRIM25 mutants Δ SPRY, 7KA and Δ CC are all impaired in IRF3 phosphorylation. However, the immunoblot shows similar levels of p-IRF3 for

7KA and Δ CC than WT, two mutants for which the interaction with G3BP1 is shown to be lost in Figure 3b. This observation undermines the general conclusion regarding the role of TRIM25 localization in SGs on the antiviral function of SGs.

Response: Thank you for your valuable comments. Given that the 7KA and Δ CC mutants were not the focus of our current study, we removed related data in the revised manuscript. In the current version, we focused TRIM25 WT, Δ PTFG and PTFG^{AAAA}, and found that TRIM25 WT significantly increased the phosphorylation level of IRF3 at Ser396 (p-IRF3) upon poly(I:C) treatment, but not Δ PTFG and PTFG^{AAAA} (Fig. 6c).

- Figure 7a-e: Reconstitutes TRIM25 knockout cells must be used for this experiment. The presence of endogenous TRIM25 makes it difficult to conclude.

Response: Thank you for your comments. Following your suggestion, we have repeated our experiments in TRIM25 KO cells (new Fig. 7a-d).

- Recent publication reports the ubiquitination of G3BP1 by TRIM25 (Yang Y, et al. 2023, Biochim Biophys Acta Gene Regul Mech. 10;1866(3):194954. doi: 10.1016/j.bbagr.2023.194954. PMID: 37302696). This new reference should be discussed.

Response: Thank you for your comments. Following your suggestion, we have cited and discussed this new study in our revised manuscript (line 503-505).

Minor comments:

- Figure 1b: add the untreated panel.

Response: We have fixed this.

- Figure 1e: label e is missing in the figure.

Response: We have fixed this.

- Figure S1: The use of the terminology WT vs Poly(I:C) is not the best appropriated, rather untreated vs poly(I:C).

Response: We appreciate the reviewer's suggestions. We have fixed this.

- Figure 2. Legend of panel 2a states that the data are presented are means \pm SEM at each time point. The panel only shows still images of the FRAP experiment, no recovery value is indicated. Naming the TRIM25-positive foci in cells "droplets" is also confusing.

Response: Thanks for the suggestion. We have modified the legend, specifically the mean \pm SEM for each time point is not shown in Fig. 2a (Line 800-801). As you suggested, we highlighted TRIM25-positive foci in the cells. Specifically, we have replaced "droplets" with "foci".

- Text line 187: TRIM25 and G3BP1 undergo dsRNA-dependent condensation: this is not correct. dsRNA promotes condensation but is dispensable.

Response: We have modified the sentence as suggested.

- Figure 7a-e, the legend states that TRIM25 PTFG-AAAA and Δ PTFG mutants were used. Only results for the mutant PTFG-AAAA are shown.

Response: Thank you for your suggestion. We have modified it. We apologize for any confusion caused by our labeling error.

Reviewer #4 (Remarks to the Author):

Overview:

In this manuscript, “TRIM25 specifies antiviral stress granules”, Shang et al. describe the co-condensation of TRIM25 and G3BP1 to stress granules in a dsRNA-dependent manner. Using a multidisciplinary approach of biochemistry, biophysics, and cell biology, the authors provide insights into how these proteins regulate the dynamics and formation of droplets in vitro and stress granules in vivo. A central conclusion of this manuscript is that through multiple distinct interactions, TRIM25 and G3BP1 promote stress granule formation to activate the RIG-I antiviral signalling pathway. While interesting, many of the results in this manuscript have been reported (Sanchez et al., 2018; Yang et al., 2022), which reduces the novelty of these findings. Another concern is that several experiments – for example, the ubiquitination assays – should have been performed more rigorously to justify the conclusions in the text (see points below). It is because of these reasons that I cannot, at this point, recommend this manuscript for publication, but I sincerely hope the authors find these comments helpful for further improving their manuscript.

Major points:

1. Line 134 & Fig. 1e, the authors should quantify the puncta/cell and display the data as a bar graph with errors.

Response: Thank you very much for your suggestion. We have quantified the puncta/cell and display the data as a scatterplot with errors (new Fig. 1i) in the revised manuscript.

2. Line 154 & Fig. S2c, the authors should quantify this difference (number & size) and display the data as a bar graph with errors.

Response: Thank you for your suggestion. We have quantified the puncta area and display the data as a violin plot form (new Fig. S2d) in the revised manuscript.

3. Fig. 3c, the authors should quantify the difference in co-condensation frequency at the 8 h time point (number & size) and display the data as a bar graph with errors.

Response: Thank you for your insight questions. We quantified the shortest distance between TRIM25 WT and Δ PTFG to G3BP1 upon poly(I:C) or sodium arsenite stimulation (Graphic 2). Unlike TRIM25 WT (Fig. 1c), TRIM25 Δ PTFG did not co-localize with G3BP1 upon poly(I:C) stimulation (new Fig. 3e). As suggested, the data is presented as a bar graph with errors (Graphic 2).

Graphic 2. Co-localization of TRIM25 and G3BP1 foci upon poly(I:C) or sodium arsenite treatment. HeLa cells expressing GFP-TRIM25 WT or Δ PTFG were treated with poly(I:C) or sodium arsenite, and then stained with an antibody against G3BP1. The shortest distance between TRIM25 foci and SGs was used for quantification.

4. Fig. 3d-e, the bottom panel of this figure is not visible; thus, the data could not be assessed.

Response: We apologize for our oversight. To better compare the difference between TRIM25 WT and Δ PTFG, we present these data together as new Fig. 3g-Fig. 3j.

As for Fig.3d, which is new Fig.3e in the revised manuscript, we used Line scans to show the related intensity profiles of TRIM25 and G3BP1.

5. Fig. 5b-c, using this approach, you cannot determine the levels of substrate ubiquitination. The authors should first enrich ubiquitin (e.g., with tandem ubiquitin-binding entities -TUBEs) and then blot for the substrates (IGF2BP2 and ZC3HAV1). This would allow for a direct readout of substrate ubiquitination levels.

Response: We followed your advice and enriched ubiquitinated proteins using tandem ubiquitin-binding entities (TUBE) (new Fig. S6c). We found that poly(I:C) stimulation increased the ubiquitination levels of RIG-I, IGF2BP2, and PABPC1, another TRIM25 substrate (ZC3HAV1 antibody was not available for immunoblotting). Importantly, the ubiquitination levels of the substrates were further increased when TRIM25 WT, but not TRIM25 Δ PTFG, was expressed. On the other hand, G3BP1 deletion diminished the substrate ubiquitination levels (new Fig. S6c). This is totally consistent with our initial conclusion (new Fig. 5b-c; new Fig. 6a-b).

6. Fig. 5d & Fig. S6a should have a negative control. The authors should spike the ITFG peptide into these reactions to determine if this interferes with TRIM25 substrate ubiquitination.

Response: Thank you for your constructive suggestion. We spiked the ITFG peptide, as suggested, and found that the ITFG peptide from SARS-CoV-2 inhibited TRIM25 ubiquitination activity (new Fig. 5d-e).

7. As RNA binding is also known to enhance TRIM25 activity (Sanchez et al. 2018), it would be interesting to compare RNA- vs. G3BP1-mediated TRIM25 activation in Fig. 5d.

Response: Thank you for your valuable comments. Our results showed that G3BP1 enhanced TRIM25 activity in the presence of dsRNA. Given that the addition of dsRNA and G3BP1 are two

distinct variables, it is difficult to make direct comparisons.

8. In the method section for the in vitro ubiquitination assays, the authors report only using the E2 enzyme Ubc13 in their assays. However, Mms2/UBE2V2 should also be included in these assays since Ubc13/Mms2 forms a heterodimer to assemble Lys63 ubiquitin chains.

Response: Thanks a lot for the suggestion. We actually used the Ubc13/UBE2V2 dimer in our experiments. We have corrected our figure (new Fig. S6d-e) and method section (line 727-734), as suggested.

9. For Fig. 6a-b, see point 5 above.

Response: Thank you for your suggestion. Please see response 5 above.

10. Lines 373-376, the authors state, “Similarly, we also found that the E3 ligase activity (TRIM25 CS), RNA binding (Δ SPRY and 7KA), and dimerization (the coiled-coil mutant Δ CC) of TRIM25 were all required to fully phosphorylate IRF3 following poly(I:C) treatment.” However, the levels of IRF3-p look the same as WT.

Response: Thank you for your valuable comments. Given that the 7KA and Δ CC mutants were not the focus of our current study, we removed related data in the revised manuscript. In the current version, we focused TRIM25 WT, Δ PTFG and PTFG^{AAAA}, and found that TRIM25 WT significantly increased the phosphorylation level of IRF3 at Ser396 (p-IRF3) upon poly(I:C) treatment, but not Δ PTFG and PTFG^{AAAA} (Fig.6c).

11. Line 376-377, the authors state, “Consistently, we also found that deletion of G3BP1 abolished TRIM25-induced p-IRF3 levels”. However, IRF3-p levels are still increasing in a TRIM25-dependent manner in the G3BP1 KO background. The authors should modify their conclusion.

Response: Thank you for these comments. We have updated our conclusions accordingly (Line 444-451). In addition, we performed quantitative RT-PCR (qPCR) analyses of IFN β and ISG56 as suggested, and found that deletion of G3BP1 also decreased poly(I:C)-induced gene expression of IFN β and ISG56 (new Fig. 6e-f).

Minor points:

1. In the figure legends, the authors should indicate how many replicates were performed for each experiment.

Response: Thanks to your suggestion, we have clearly stated that three replicates of each experiment were performed in the revised manuscript.

2. In Fig. 2a legend, the authors state, “Data are presented as mean plus/minus standard error of the mean (s.e.m.) at each time point from three independent replicates.” However, there is no data presented this way in the figure.

Response: We apologize for not explaining this point clearly in our original submission. Descriptions have been updated in the text and figure legends (line 800-801).

3. Line 110-111, the authors state, “Thus, poly(I:C) treatment stimulates the recruitment of TRIM25

into the SGs”; however, this is not shown until the following figure (Fig. 1c).

Response: Following your suggestion, we have modified the conclusion accordingly (line 114-115).

4. Fig. 1e, the “e” is missing in the figure.

Response: We apologize for the oversight. It has been corrected in the revised manuscript.

5. In Fig. 3c, the labels within the figure must be clearly visible (i.e., text in left panels).

Response: Thank you for your suggestion. A larger field of view and clear labels are provided in the revised manuscript (Fig. 3d).

6. In the WT vs. Δ PTFG interactome study, was this experiment performed in cells treated with poly(I:C)?

Response: Thank you for your question. In the WT vs. Δ PTFG interactome study, this experiment was performed in cells treated without poly(I:C).

7. Fig. 7a-d, statistical analysis should be performed as in Fig. 6e.

Response: Thank you for your suggestion. We have revised as suggested.

Reviewers' Comments:

Reviewer #1:

Remarks to the Author:

The authors have properly addressed all the concerns raised by the reviewer. The data quality presented in the paper is sufficient to support the main points. The study provides a comprehensive description of TRIM25 as a primary marker protein that responds to virus infections and forms virus-specific stress granules in conjunction with G3BP1. This detailed investigation into stress granules contributes to a better understanding of their diversity and functional roles. Considering the quality of the work, I believe it is suitable for publication in Nature Communications.

The title is now revised as "TRIM25 predominantly associates with anti-viral stress granules,". If the authors wish to reconsider the title, they could focus on highlighting the specific findings or the unique characteristics of TRIM25 in relation to stress granules. Here are a few alternative title suggestions:

"TRIM25-mediated formation of virus-specific stress granules during antiviral responses"
"Investigating the role of TRIM25 in the formation and function of antiviral stress granules"

Reviewer #2:

Remarks to the Author:

This reviewer has no further comments.

Reviewer #3:

Remarks to the Author:

Shang and colleagues provided comprehensive answers to my questions. Thanks to this additional work, all the concerns I still had were addressed, which greatly improved the overall quality of the manuscript. I did note that the authors inadvertently provided the wrong new figure 3 (which appears to be a duplicate of figure 4). Although this does not directly affect the answers to my questions, the original figure will have to be resubmitted to complete the revision. I am confident that after this correction, the results of this research will be of great value to the field.

Reviewer #4:

Remarks to the Author:

In the resubmitted manuscript, 'TRIM25 predominately associates with antiviral stress granules', by Da Jia and colleagues, the authors have significantly improved their manuscript on several fronts, especially regarding the quantitative analysis of stress granule formation in multiple figures. However, I still have concerns about the quality of the biochemical characterisation of the G3BP1-TRIM25 association using ubiquitination assays and the interpretation of these results (see points below). Moreover, the data from Figure 3 is missing from the manuscript file, and the data from Figure 4 is shown twice (both as Figure 3 and Figure 4). This omission makes the assessment of the entire manuscript impossible. For these reasons, I cannot recommend this manuscript for publication, but I sincerely hope the authors find the comments below helpful.

Major points:

- 1) Lines 360 – 367, Fig. S6b; In the comparison of poly(I:C) vs sodium arsenite IGF2BP2 ubiquitination, the authors state that poly(I:C) treatment led to higher levels of IGF2BP2 ubiquitination. However, to this reviewer, the ubiquitination levels between these two conditions look nearly identical.
- 2) Line 378 – 384, Fig. S6c: To assess the levels of substrate ubiquitination in TUBE pull-down

assays, the entire molecular weight range above each substrate should be shown. The blots should not be cropped as shown in Fig. S6c.

3) Line 393 – 394, Fig. S6d: In the in vitro ubiquitination assays with increasing concentrations of G3BP1 WT, the authors state that G3BP1 further enhanced the production of polyUb chains. In the UbcH5b/TRIM25 reactions, I can somewhat understand how the authors came to this conclusion. However, the problem is that increasing the concentration of G3BP1 also increases the concentration of Lysine residues that can be ubiquitinated. Therefore, in the current assay format, it is difficult to distinguish between enhanced TRIM25 activity due to co-condensation and elevated Ub modifications due to increased Lysine availability. The authors should repeat these experiments with a G3BP1 mutant that is defective TRIM25 binding (e.g., the G3BP1 TBM mutant). In the Ubc13/UBE2V2/TRIM25 reactions, the levels of ubiquitination are very similar, and so to this reviewer, the result does not agree with the authors' interpretation of the data. One other point – the G3BP1 concentration is shown as mM in the figure, which is likely a typo.

4) Line 394 – 397, Fig. S6e: In the in vitro ubiquitination assays comparing the impact of dsRNA vs dsRNATATA on TRIM25 E3 ligase activity, the ubiquitin levels appear very similar, especially in the Ubc13/UBE2V2 reactions. Therefore, this reviewer does not agree with the authors' interpretation of the data. The authors should perform a time course analysis on these assembly reactions to further characterise the impact of these different dsRNA molecules on TRIM25 activity.

5) Line 399 – 408, Fig. 5d-e: In the ubiquitination assays comparing RIG-I-2CARD and ZC3HAV1 substrates, the impact of the ITFG peptide on substrate ubiquitination appears minimal, especially with regards to the ZC3HAV1 substrate. The authors should perform a time course analysis on these assembly reactions to further characterize the impact of the ITFG peptide on substrate Ub levels. Also, it is unclear from the legend or methods section what concentration of ITFG peptide was used in these assays.

6) Line 424 – 427, Fig. 5g-h; In the ubiquitination assays comparing RIG-I-2CARD and ZC3HAV1 substrates in the presence of WT G3BP1 and mutants, the impact of the mutants on substrate ubiquitination levels appears negligible and therefore this data does not agree with the authors' conclusions that G3BP1 dimerization and G3BP1-TRIM25 interactions enhance TRIM25 activity. This result/figure is confusing since a similar experiment in the original manuscript looked much more promising. Could the authors please explain this discrepancy?

Point-by-point responses:

We appreciate the reviewers for their constructive suggestions, which were very helpful in improving our manuscript. We have updated the data in Figure 3 and apologize for the error. In response to the issues raised by the reviewers, we have corrected the data for the ubiquitination assays section as well as added key experimental variables. Our point-by-point response is provided below:

REVIEWER COMMENTS:

Reviewer #1 (Remarks to the Author):

The authors have properly addressed all the concerns raised by the reviewer. The data quality presented in the paper is sufficient to support the main points. The study provides a comprehensive description of TRIM25 as a primary marker protein that responds to virus infections and forms virus-specific stress granules in conjunction with G3BP1. This detailed investigation into stress granules contributes to a better understanding of their diversity and functional roles. Considering the quality of the work, I believe it is suitable for publication in Nature Communications.

The title is now revised as "TRIM25 predominantly associates with anti-viral stress granules,". If the authors wish to reconsider the title, they could focus on highlighting the specific findings or the unique characteristics of TRIM25 in relation to stress granules. Here are a few alternative title suggestions:

"TRIM25-mediated formation of virus-specific stress granules during antiviral responses"

"Investigating the role of TRIM25 in the formation and function of antiviral stress granules"

Response: Thanks a lot for the suggestions. However, the two suggested titles did not cover the major discoveries of our studies. Thus, we stick to the title that we used.

Reviewer #2 (Remarks to the Author):

This reviewer has no further comments.

Reviewer #3 (Remarks to the Author):

Shang and colleagues provided comprehensive answers to my questions. Thanks to this additional work, all the concerns I still had were addressed, which greatly improved the overall quality of the manuscript. I did note that the authors inadvertently provided the wrong new figure 3 (which appears to be a duplicate of figure 4). Although this does not directly affect the answers to my questions, the original figure will have to be resubmitted to complete the revision. I am confident that after this correction, the results of this research will be of great value to the field.

Response: We apologize for the oversight. In the revised manuscript, the correct figure is provided.

Reviewer #4 (Remarks to the Author):

In the resubmitted manuscript, 'TRIM25 predominately associates with antiviral stress granules', by Da Jia and colleagues, the authors have significantly improved their manuscript on several fronts, especially regarding the quantitative analysis of stress granule formation in multiple figures. However, I still have concerns about the quality of the biochemical characterization of the G3BP1-TRIM25 association using ubiquitination assays and the interpretation of these results (see points below). Moreover, the data from Figure 3 is missing from the manuscript file, and the data from Figure 4 is shown twice (both as Figure 3 and Figure 4). This omission makes the assessment of the entire manuscript impossible. For these reasons, I cannot recommend this manuscript for publication, but I sincerely hope the authors find the comments below helpful.

Major points:

1) Lines 360 – 367, Fig. S6b; In the comparison of poly(I:C) vs sodium arsenite IGF2BP2 ubiquitination, the authors state that poly(I:C) treatment led to higher levels of IGF2BP2 ubiquitination. However, to this reviewer, the ubiquitination levels between these two conditions look nearly identical.

Response: We appreciate the reviewer's thoughtful suggestion. The issue is likely due to blot exposure time - longer exposure minimizes the difference. In the revised manuscript, these images were replaced by that of shorter exposure and we also quantified the difference from multiple experiments (New. Fig. S6b).

2) Line 378 – 384, Fig. S6c: To assess the levels of substrate ubiquitination in TUBE pull-down assays, the entire molecular weight range above each substrate should be shown. The blots should not be cropped as shown in Fig. S6c.

Graphic 1 TUBEs pull down ubiquitinated substrates

Response: We thank the reviewers for this comment. Images showing the molecular weight range above the substrate are shown in Graphic 1, which are nearly identical to those shown in Figure S6c.

3) Line 393 – 394, Fig. S6d: In the in vitro ubiquitination assays with increasing concentrations of G3BP1 WT, the authors state that G3BP1 further enhanced the production of polyUb chains. In the UbcH5b/TRIM25 reactions, I can somewhat understand how the authors came to this conclusion. However, the problem is that increasing the concentration of G3BP1 also increases the concentration of Lysine residues that can be ubiquitinated. Therefore, in the current assay format, it is difficult to distinguish between enhanced TRIM25 activity due to co-condensation and elevated Ub modifications due to increased Lysine availability. The authors should repeat these experiments with a G3BP1 mutant that is defective TRIM25 binding (e.g., the G3BP1 TBM mutant). In the Ubc13/UBE2V2/TRIM25 reactions, the levels of ubiquitination are very similar, and so to this reviewer, the result does not agree with the authors' interpretation of the data. One other point – the G3BP1 concentration is shown as mM in the figure, which is likely a typo.

Response: We are very grateful for your suggestions, and have performed new experiment with G3BP1 TBM mutant (New. Fig. S6d). Unlike G3BP1 WT, G3BP1 TBM did not further enhanced TRIM25 activity (New. Fig. S6d). We also correct the labeling of G3BP1 concentration as suggested (New. Fig. S6d).

Unlike UbcH5b, Ubc13/UBE2V2 does not covalently attach Ub to the substrate (PMD: 27425606). Through our experiments, we have obtained more consistent results with UbcH5b than that of Ubc13/UBE2V2. Since the Ubc13/UBE2V2 E2 ligase is the main focus of our study, we decided to show results of UbcH5b only.

4) Line 394 – 397, Fig. S6e: In the in vitro ubiquitination assays comparing the impact of dsRNA vs dsRNA^{TATA} on TRIM25 E3 ligase activity, the ubiquitin levels appear very similar, especially in the Ubc13/UBE2V2 reactions. Therefore, this reviewer does not agree with the authors'

interpretation of the data. The authors should perform a time course analysis on these assembly reactions to further characterize the impact of these different dsRNA molecules on TRIM25 activity.

Graphic 2 Time course analysis of in vitro ubiquitination. Assays contained purified E1, E2 (UbcH5b), E3 (TRIM25), Ub, and dsRNA (56bp-dsRNA or 56bp-dsRNA^{TATA}). Poly-Ub chain was analyzed by anti-Ub blot.

Response: We thank the reviewer for the suggestions, and performed time course analysis of in vitro ubiquitination (Graphic 2). The addition of 56bp dsRNA stimulated TRIM25 activity more dramatically than that of 56bp dsRNA^{TATA} (Graphic 2).

5) Line 399 – 408, Fig. 5d-e: In the ubiquitination assays comparing RIG-I-2CARD and ZC3HAV1 substrates, the impact of the ITFG peptide on substrate ubiquitination appears minimal, especially with regards to the ZC3HAV1 substrate. The authors should perform a time course analysis on these assembly reactions to further characterize the impact of the ITFG peptide on substrate Ub levels. Also, it is unclear from the legend or methods section what concentration of ITFG peptide was used in these assays.

Graphic 3 Time course analysis of in vitro ubiquitination. Time course analysis of in vitro ubiquitination analysis of ZC3HAV1-N-HA. Assays contained E1 ligase, UbcH5b (E2 ligase), TRIM25 (E3 ligase), the 56-bp dsRNA, Ub, and ZC3HAV1-N-HA. Ubiquitination of ZC3HAV1-N-HA was analyzed by anti-HA blot.

Response: We thank the reviewers for the suggestions. The issue is likely due to blot exposure

time and/or image formatting. The figures have been updated in the revised manuscript (Fig. 5d-e).

At the same time, we also performed a time-course analysis of ubiquitination assay following your suggestion. The addition of ITFG peptide significantly inhibited the ubiquitination of ZC3HAV1-N by TRIM25 (Graphic 3). The concentration of ITFG peptide used has been added to the figure legends for Fig. 5d-e.

6) Line 424 – 427, Fig. 5g-h; In the ubiquitination assays comparing RIG-I-2CARD and ZC3HAV1 substrates in the presence of WT G3BP1 and mutants, the impact of the mutants on substrate ubiquitination levels appears negligible and therefore this data does not agree with the authors' conclusions that G3BP1 dimerization and G3BP1-TRIM25 interactions enhance TRIM25 activity. This result/figure is confusing since a similar experiment in the original manuscript looked much more promising. Could the authors please explain this discrepancy?

Response: We thank the reviewer for the comments. The issue is likely due to blot exposure time and/or image formatting. The figures have been updated in the revised manuscript (Fig. 5g-h).

The difference is likely due to different E2 enzymes and different batches of proteins. Unlike UbcH5b, Ubc13/UBE2V2 does not covalently attach Ub to the substrate (PMD: 27425606). Through our experiments, we have obtained more consistent results with UbcH5b than that of Ubc13/UBE2V2. Since the Ubc13/UBE2V2 E2 ligase is the main focus of our study, we decided to show results of UbcH5b only.

Reviewers' Comments:

Reviewer #4:

Remarks to the Author:

This reviewer has no additional comments.